# MIN-K%++: IMPROVED BASELINE FOR DETECTING PRE-TRAINING DATA FROM LARGE LANGUAGE MODELS

**Jingyang Zhang**[1]*, **Jingwei Sun**[1]*, **Eric Yeats**[1], **Yang Ouyang**[1], **Martin Kuo**[1],
**Jianyi Zhang**[1], **Hao Frank Yang**[1,2], **Hai Li**[1]
[1]Duke University    [2]Johns Hopkins University
`https://zjysteven.github.io/mink-plus-plus/`

## ABSTRACT

The problem of pre-training data detection for large language models (LLMs) has received growing attention due to its implications in critical issues like copyright violation and test data contamination. Despite improved performance, existing methods are mostly developed upon simple heuristics and lack solid, reasonable foundations. In this work, we propose a novel and theoretically motivated methodology for pre-training data detection, named Min-K%++. Specifically, we present a key insight that training samples tend to be local maxima of the modeled distribution along each input dimension through maximum likelihood training, which in turn allow us to insightfully translate the problem into identification of local maxima. Then, we design our method accordingly that works under the discrete distribution modeled by LLMs, whose core idea is to determine whether the input forms a mode or has relatively high probability under the conditional categorical distribution. Empirically, the proposed method achieves new SOTA performance across multiple settings (evaluated with 5 families of 10 models and 2 benchmarks). On the WikiMIA benchmark, Min-K%++ outperforms the runner-up by 6.2% to 10.5% in detection AUROC averaged over five models. On the more challenging MIMIR benchmark, it consistently improves upon reference-free methods while performing on par with reference-based method that requires an extra reference model.

## 1 INTRODUCTION

Data is one of the most important factors for the success of large language models (LLMs). As the training corpus grows in scale, it has increasing tendency to be held in-house as proprietary data instead of being publicly disclosed (Touvron et al., 2023b; Achiam et al., 2023). However, for large-scale training corpora that consist of up to *trillions* of tokens (Computer, 2023), the sheer volume of the training corpus can lead to unintended negative consequences. For example, memorized private information is vulnerable to data extraction (Carlini et al., 2021), and memorized copyrighted contents (*e.g.*, books and news articles) may violate the rights of content creators (Grynbaum & Mac, 2023; Knibbs, 2023). Furthermore, it becomes increasingly likely that evaluation data is exposed at training time, bringing the faithfulness and effectiveness of evaluation benchmarks into question (Oren et al., 2023).

For these reasons, there has been growing interest in effective *pre-training data detection* strategies. Pre-training data detection can be considered a special case of Membership Inference Attack (MIA) (Shokri et al., 2017): the goal is to infer whether a given input has been used for training a target LLM (see Figure 1 left for illustration). Due to characteristics of pre-training corpora and training characteristics of LLMs (Shi et al., 2024; Duan et al., 2024), this problem has been shown to be much more challenging than conventional MIA settings (see Section 2 for details). There are a few methods proposed recently, dedicated to this problem (Carlini et al., 2021; Mattern et al., 2023; Shi

---

*Equal contribution.

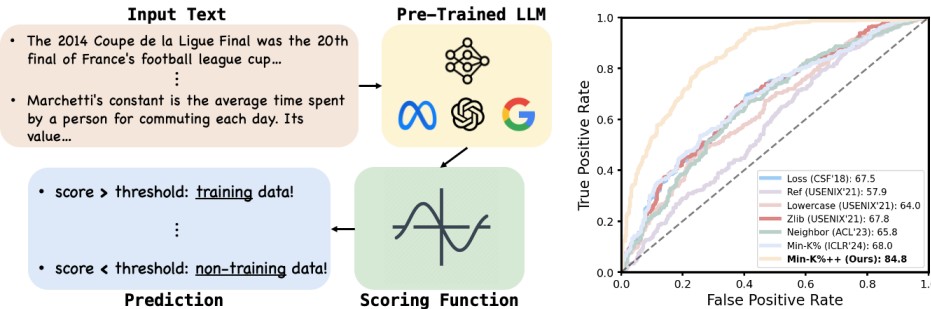

Figure 1: **Left:** We study the pre-training data detection problem for LLMs (Shi et al., 2024; Duan et al., 2024). Given the grey-box access to the target LLM (only the logits, token probabilities, and loss are available), the goal is to design a scoring function that yields a score which (upon thresholding) best separates training and non-training text. **Right:** ROC (receiver operating characteristic) curves of various methods on WikiMIA benchmark (Shi et al., 2024). The AUROC (area under the ROC curve) score is denoted in the legend. Our Min-K%++ improves upon existing approaches by large margin.

et al., 2024). However, despite their improved performance, most existing methods are developed with simple heuristics, lacking solid and interpretable foundations.

In this work, we propose a novel and theoretically motivated methodology, Min-K%++, for pre-training data detection. Our exploration starts by asking a fundamental question of "what characteristics or footprints do training samples exhibit within the model", where we answer it by revisiting the maximum likelihood training objective through the lens of score matching (Hyvärinen & Dayan, 2005). Importantly, we uncover that for continuous distributions with maximum likelihood training, training data points tend to be local maxima or be located near local maxima along input dimensions, a key insight that allows us for the first time to translate the training data detection problem into identification of local maxima. Then, upon theoretical insight, we develop a practical scoring mechanism that works for the discrete distribution modeled by LLMs. The core idea is to examine whether the input next token forms a mode or has relatively high probability under the conditional categorical distribution.

Empirically, we perform extensive experiments to verify the proposed method on two established benchmarks with up to 5 families of models, including LLaMA (Touvron et al., 2023a), Pythia (Biderman et al., 2023), and the new state-space architecture Mamba (Gu & Dao, 2023). Figure 1 right showcases the performance comparison between our method and previous ones. Notably, Min-K%++ on average leads to 6.2% to 10.5% absolute increases over the runner-up Min-K% (Shi et al., 2024) on the WikiMIA benchmark (Shi et al., 2024). On the more challenging MIMIR benchmark, our method still consistently outperforms other *reference-free* methods and is on par with the reference-based method (Carlini et al., 2021): unlike reference-based methods which require another LLM to calibrate the likelihood, Min-K%++ is applied to a standalone target LLM. We also study an online detection setting that simulates "detect-while-generating"; the proposed Min-K%++ again performs the best. Lastly, Min-K%++ can also be interpreted from a calibration perspective: with ablation study we show that both calibration factors in the formulation contribute to the overall high performance. We summarize our contributions as follows:

**(1)** We establish that for maximum likelihood training, training instances tend to form local maxima or locate near local maxima along each input dimension (under continuous distributions), allowing us to turn training data detection into local maxima identification. **(2)** We develop a novel and sound methodology for LLM training data detection which functions by examining whether the input forms a mode or has relatively high probability under the conditional categorical distribution. **(3)** We demonstrate remarkable improvements over existing methods on established benchmarks.

## 2 RELATED WORK

**Membership Inference Attacks.** MIA has long been a security and privacy-related topic. Given a target model and a target input, MIA aims to infer whether the input has been used to train the

target model (Shokri et al., 2017; Yeom et al., 2018). It has been extensively studied in both vision (Carlini et al., 2022a;b; 2023b; Zarifzadeh et al., 2023) and language domains (Carlini et al., 2021; Watson et al., 2022; Mattern et al., 2023). Besides investigations on advanced methodologies, MIA also has profound implications in quantifying privacy risks (Mireshghallah et al., 2022a), measuring memorization (Carlini et al., 2023a), helping with privacy auditing (Steinke et al., 2023; Yao et al., 2024), and detecting test-set contamination (Oren et al., 2023) and copyrighted contents (Meeus et al., 2023; Duarte et al., 2024). In particular, benchmark leakage has been a serious problem in the LLM era, and researchers have dedicated endeavors towards assessing and/or mitigating it (Xu et al., 2024; Zhang et al., 2024).

**Pre-training data detection for LLMs.** Despite being an instance of MIA (the general definition and evaluation metrics remain the same), this problem poses unique challenges compared with conventional MIA settings (Shi et al., 2024; Duan et al., 2024). For example, many early MIA methods need to train shadow models on the same distribution as the target model, which is no longer practical as many LLMs' pre-training corpus is nonpublic. Second, characteristics of LLM pre-training (*e.g.*, few training epochs, large-scale training) inherently makes MIA much more challenging. Lastly, most existing works on MIA against LLMs target the fine-tuning stage (Mireshghallah et al., 2022b; Fu et al., 2023; Mattern et al., 2023), which cannot transfer to the pre-training stage for the same reasons.

As a result, there has been growing interests in pre-training data detection recently, though it is still largely underexplored. Shi et al. (2024), to our knowledge, is the first to investigate this problem. They contribute the WikiMIA benchmark and propose the Min-K% method. Duan et al. (2024) perform systematic evaluation with the constructed MIMIR benchmark and analyze the challenges. There are also works that specifically investigate copy-righted content detection, *e.g.*, identifying memorized books (Duarte et al., 2024). In this work, we propose a novel method that outperforms the previously best-performing Min-K% and achieve superior performances over existing (reference-free) methods on both benchmarks. In addition, our insightful finding of that training data tends to form local maximum is unique, which relates to but complements and solidifies previous intuitions that LLM's output distribution is most likely to be peaked when it is emitting memorized data (Dong et al., 2024).

## 3 BACKGROUND

In this section, we first cover the problem statement of pre-training data detection defined by prior works (Shokri et al., 2017; Shi et al., 2024; Duan et al., 2024). Then we briefly introduce how LLMs work and how they are trained.

**Problem statement.** Pre-training data detection is an instance of Membership Inference Attack (MIA) (Shokri et al., 2017). Formally, given 1) a data instance $x$ and 2) a pre-trained auto-regressive LLM $\mathcal{M}$ that is trained on a dataset $\mathcal{D}$, the goal is to infer whether $x \in \mathcal{D}$ or not (*i.e.*, $x$ is training data or non-training data). The approach to detection leverages a scoring function $s(x; \mathcal{M})$ that computes a score for each input. A threshold is then applied to the score to yield a binary prediction:

$$\text{prediction}(x, \mathcal{M}) = \begin{cases} 1 \ (x \in \mathcal{D}), & s(x; \mathcal{M}) \geq \lambda \\ 0 \ (x \notin \mathcal{D}), & s(x; \mathcal{M}) < \lambda \end{cases}, \tag{1}$$

where $\lambda$ is a case-dependent threshold. Following the established standard (Mattern et al., 2023; Shi et al., 2024; Duan et al., 2024), we consider *grey-box* access to the target model $\mathcal{M}$, meaning that one can only access the output statistics including the loss value, logits, and token probabilities. Additional information such as the model weights and gradients are not available. In summary, the key to the pre-training data detection is to design an appropriate scoring function that best separates training data from non-training data.

**(Auto-regressive) LLMs.** LLMs are typically trained by maximum likelihood training that maximizes the probability of training token sequences (Radford et al., 2019; Brown et al., 2020). In particular, auto-regressive LLMs decompose the probability of token sequence $(x_1, x_2, ..., x_t)$ with chain rule (Wikipedia, 2023), *i.e.*, $p(x_1, x_2, ..., x_t) = p(x_t | x_1, x_2, ...x_{t-1}) \cdot p(x_1, x_2, ...x_{t-1})$. For brevity, throughout the paper we abbreviate the prefix of $x_t$ as $x_{<t}$. At inference stage, LLMs yield new tokens one by one according to its predicted conditional categorical distribution $p(\cdot | x_{<t})$ over the vocabulary.

## 4 MIN-K%++

In this section we introduce our methodology, called Min-K%++. We start by discussing our grounding observation around general maximum likelihood training for continuous distribution, where we unveil that training samples tend to form local maxima or locate near local maxima along each input dimension of the distribution captured by the model. This insight allows us to introduce a unique and theoretically sound perspective for addressing the problem. Then, translating our insight into a practical solution, we formulate the proposed Min-K%++ which works for discrete inputs of LLMs by examining whether the input forms a mode or has a relatively high probability under the conditional categorical distribution of the predicted next token.

### 4.1 MOTIVATION & INSIGHT

To approach the pre-training data detection problem, unlike prior methods that often rely on heuristics (Yeom et al., 2018; Carlini et al., 2022a; Shi et al., 2024), we choose to start our in-depth exploration by asking a fundamental question that existing works fail to touch to our knowledge. That is, *what characteristics or "footprints" do training samples leave to the model after training?* Since training data interacts with the model through training-time optimization, to answer the question, we revisit the maximum likelihood training objective of (auto-regressive) LLMs through the lens of score matching (Hyvärinen & Dayan, 2005). For the ease of discussion, let us generalize from LLMs, which model discrete distribution over tokens, to the general maximum likelihood training with continuous distribution for now.

Earlier, Koehler et al. (2022) have proved that implicit score matching (ISM) objective (Hyvärinen & Dayan, 2005) is a relaxation (within a multiplicative factor) of maximum likelihood estimation. Consequently, one can reformulate the maximum likelihood training loss with ISM as

$$\frac{1}{N} \sum_{\boldsymbol{x}} \left[ \frac{1}{2} ||\psi(\boldsymbol{x})||^2 + \underbrace{\sum_{i=1}^{d} \frac{\partial \psi_i(\boldsymbol{x})}{\partial \boldsymbol{x}_i}}_{\substack{\text{the sum of the second-order} \\ \text{partial derivatives}}} \right], \tag{2}$$

where $\psi(\boldsymbol{x}) = \frac{\partial \log p(\boldsymbol{x})}{\partial \boldsymbol{x}}$ is the score function (Hyvärinen & Dayan, 2005) for a $d$-dimensional input $\boldsymbol{x}$, $\boldsymbol{x}_i$ is the element at the $i$-th dimension of $\boldsymbol{x}$, $\psi_i(\boldsymbol{x}) = \frac{\partial \log p(\boldsymbol{x})}{\partial \boldsymbol{x}_i}$, and $N$ is the number of training samples. Without ambiguity, we are omitting the model parameters in the formulation of $\psi$ for brevity.

**Remark.** It can be seen from Equation (2) that maximum likelihood training implicitly minimizes both 1) the magnitude of first-order derivatives of likelihood $\log p(\boldsymbol{x})$ w.r.t. $\boldsymbol{x}$ and 2) the sum of the second-order partial derivatives of $\log p(\boldsymbol{x})$ w.r.t. each dimension of $\boldsymbol{x}$. As a result, we posit that the first-order $\frac{\partial \log p(\boldsymbol{x})}{\partial \boldsymbol{x}_i}$ will be close to 0 and the second-order $\frac{\partial^2 \log p(\boldsymbol{x})}{\partial \boldsymbol{x}_i^2}$ will be minimized to be negative for training samples. Importantly, according to the univariate second-derivative test (Wainwright et al., 2005), the implication here is that when looking along each dimension of the input (corresponding to each $\boldsymbol{x}_i$), training data points tend to form local maxima or be located near local maxima of the likelihood. Given this key insight, *we can translate the problem of detecting training data into identifying whether the input is a local maximum or locates near a local maximum (along input dimensions)*, providing a novel and theoretically grounded perspective for tackling this task.

**Empirical evidence.** It turns out that we can empirically verify our theoretical insight in the continuous space by considering diffusion model for image generative modeling, as 1) pixel values (after preprocessing) are continuous, and 2) diffusion models are trained by maximizing the (variational lower bound of) log likelihood (Ho et al., 2020). Here we test whether the input is a local maximum along each input dimension by perturbing random pixels of the image (one pixel corresponds to one dimension of the input) and comparing the likelihood of the original image with that of the perturbed one[1]. The result is shown in Figure 2. We see that the training image (with unperturbed pixel) is

---

[1]The ideal way of testing whether the training images form local maxima along each input dimension would be to compute the first and second-order partial derivative of log likelihood w.r.t. each pixel. However, this would be computationally infeasible as evaluating likelihood with diffusion model involves a series of chained forward passes with thousands of steps (the computation graph would be too large to compute the gradients).

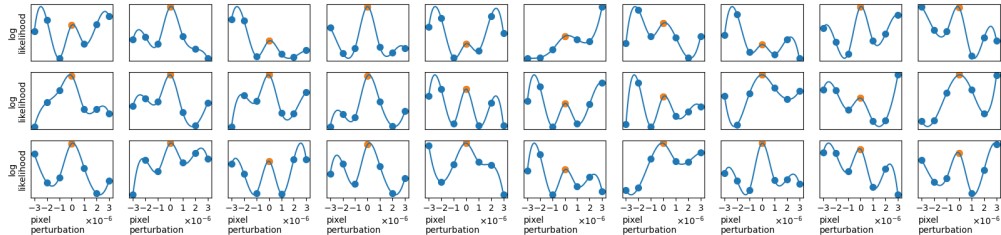

Figure 2: In each plot, we perturb a random pixel of a random training image of a diffusion model trained on CIFAR-10 and evaluate the (variational lower bound of) log likelihood. Perturbing a certain pixel is essentially moving the input along a certain dimension. We mark the original pixel with orange and the perturbed pixels with blue. We see that the training input (with the unperturbed pixel) is indeed the local maximum (the concave pattern), effectively echoing our theoretical discussion.

indeed the local maximum within its vicinity. This experiment thus verifies our theoretical insight, and actually hints at the potential of applying our idea to other modalities and models beyond LLMs.

## 4.2 FORMULATING MIN-K%++

**From insights to an actual method.** Our insight essentially indicates that a training input tends to have higher probability than other neighbor input values along each input dimension. In the context of LLMs, each dimension of the token sequence corresponds to a token. Therefore, translating the insight to the discrete categorical distribution of LLMs, we posit that each training token will tend to have higher probability relative to many other candidate tokens in the vocabulary, or even form a mode of the conditional distribution modeled by the LLM.

To achieve this, the core idea of our actual method is to compare the probability of the target token with the expected probability of all tokens within the vocabulary. Concretely, we propose the following formulation for our Min-K%++:

$$\text{Min-K\%++}_{\text{token seq.}}(x_{<t}, x_t) = \frac{\log p(x_t|x_{<t}) - \mu_{\cdot|x_{<t}}}{\sigma_{\cdot|x_{<t}}}, \tag{3}$$

$$\text{Min-K\%++}(x) = \frac{1}{|\text{min-}k\%|} \sum_{(x_{<t}, x_t) \in \text{min-}k\%} \text{Min-K\%++}_{\text{token seq.}}(x_{<t}, x_t). \tag{4}$$

Here, $\mu_{\cdot|x_{<t}} = \mathbb{E}_{z \sim p(\cdot|x_{<t})}[\log p(z|x_{<t})]$ is the expectation of the next token's log probability over the vocabulary of the model given the prefix $x_{<t}$, and $\sigma_{\cdot|x_{<t}} = \sqrt{\mathbb{E}_{z \sim p(\cdot|x_{<t})}[(\log p(z|x_{<t}) - \mu_{\cdot|x_{<t}})^2]}$ is the standard deviation. In practice, both terms can be computed analytically since the categorical distribution $p(\cdot|x_{<t})$ is encoded by the output logits of the model, which we have access to. The computation of Min-K%++ incurs no computational overhead on top of LLM inference, as we illustrate in Appendix A.

Upon the sequence-wise score computed by Equation (3), we adopt a similar strategy to Shi et al. (2024) where we select the $k\%$ of the token sequences with the minimum score and compute the average over them to obtain the final sentence-wise score, as described by Equation (4). This is because in practice LLM training minimizes the loss of all possible sequences in each training text/sentence. Therefore, by computing an aggregated score we expect to get a more robust estimation for the full sentence input. While our method is closely motivated by our theoretical insight, below we provide further interpretations for more comprehensive understanding.

**Interpretation 1: More robust identification of the mode.** We have translated training data detection into determining whether the input token has high probability or even forms a mode of the conditional distribution. Besides motivating our method, this unique scheme also allows us to understand why existing methods may be sub-optimal (which was unclear earlier since they were mostly developed upon heuristics).

Specifically, let us take the state-of-the-art Min-K% (Shi et al., 2024) as an example, which uses the token probability as the score based on a simple intuition that training texts are less likely to include low-probability "outlier" tokens. This actually may not identify the mode robustly or accurately as

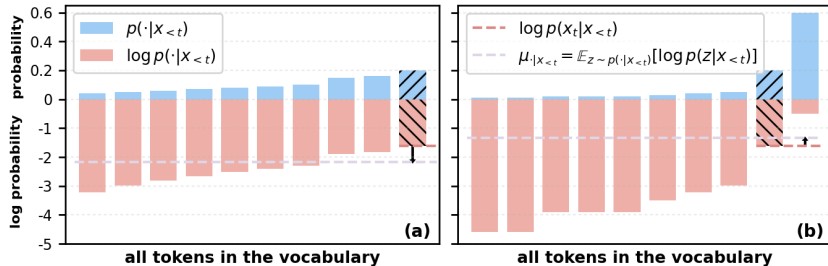

Figure 3: A conceptual example showcasing the idea of Min-K%++. (a) and (b) show two next token distributions for two different input token sequences $(x_{<t}, x_t)$. The bars with the hatch pattern ('//') correspond to the target next token $x_t$. Existing methods often measure the exact next token probability of $x_t$ (hatched bars), which fails to separate these two inputs due to the same probability (0.2). By comparison, Min-K%++ determines if $x_t$ forms a mode or has relatively high probability by comparing $\log p(x_t|x_{<t})$ with $\mu_{\cdot|x_{<t}}$; the score is the difference between the red and pink dashed lines. See text for detailed discussion.

token probabilities can vary a lot across different inputs. For example, some training texts may be rare or inherently difficult to learn[2], thus having low token probabilities despite still being the mode. Our design, in contrast, can account for such cases since it explicitly compares the log likelihood of the target next token $\log p(x_t|x_{<t})$ and the expected log likelihood over all candidate tokens $\mu_{\cdot|x_{<t}}$. Regardless of the absolute value of the token likelihood, the larger $\log p(x_t|x_{<t}) - \mu_{\cdot|x_{<t}}$ is, the more likely $x_t$ forms a mode or has a relatively high probability among the vocabulary.

To see this more clearly, let us consider a conceptual example depicted in Figure 3, where we show the predicted next token probability distribution for two different input token sequences $(x_{<t}, x_t)$, supposing a vocabulary of size 10. In the two cases (Figure 3 (a) and (b)), both $p(x_t|x_{<t})$ is 0.2. Existing methods often measure exactly the next token probability (Shi et al., 2024) and thus will give the same judgement to both inputs (*e.g.*, that the sequence $(x_{<t}, x_t)$ was not seen in the training). However, inspecting them with the idea of our Min-K%++ reveals their distinct characteristics.

In Figure 3 (a), one can notice that the distribution is actually peaked at $x_t$ (the bar with the hatch pattern '//'), indicating that the current sequence $x_t$ forms a mode of $p(\cdot|x_{<t})$ and is very likely to have appeared in the training. In Figure 3 (b), the probability distribution is not peaked at $x_t$ as there is another token that gets assigned much larger probability mass. This means that $x_t$ is less likely to follow the prefix $x_{<t}$ according to the learned distribution of the model, which should be seen as an indicator that the sequence $(x_{<t}, x_t)$ was not seen in the training.

The above reasoning can be concisely reflected by comparing $\log p(x_t|x_{<t})$ and $\mu_{\cdot|x_{<t}}$, which are marked by the two dashed lines in Figure 3. Accordingly, it can be seen that Figure 3 (a) and (b) exhibit positive and negative $\log p(x_t|x_{<t}) - \mu_{\cdot|x_{<t}}$, respectively, which successfully separate them apart.

**Interpretation 2: A calibration perspective.** Reference-based membership inference attacks (Carlini et al., 2021) show superior performance by calibrating the (sentence-level) likelihood with certain "references", such as the likelihood of the same input on a smaller LLM, the Zlib entropy, or the likelihood of lowercased text. In this regard, Min-K%++ can be seen as a calibrating method, too, where it calibrates the next token log likelihood with two calibration factors, $\mu_{\cdot|x_{<t}}$ and $\sigma_{\cdot|x_{<t}}$. Our method is unique in that instead of using some external references (*e.g.*, an extra LLM), we leverage the statistics that can be readily computed with the target model only.

Concretely, the effect of $\mu_{\cdot|x_{<t}}$ has been thoroughly discussed in the first interpretation. The inclusion of $\sigma_{\cdot|x_{\le t}}$ is inspired by temperature scaling (Guo et al., 2017), a technique for calibrating the prediction probability of neural networks. It scales the model output by a constant and has been shown to benefit tasks like out-of-distribution detection (Liang et al., 2018; Zhang et al., 2023). Such idea is well-suited for our task and method as we are closely looking at the the predicted

---

[2]We note that rare and difficult-to-learning training texts could be challenging for all methods and needs more research efforts in the future. Here we use such scenario only to exemplify the advantage of Min-K%++ over existing methods.

probabilities of LLMs. In the meantime, while temperature scaling uses a constant scaling factor, we posit that the suitable temperature can vary across inputs and models for training data detection. Therefore, we intend to have a dynamic scaling factor that is adaptive to different cases, which would make the calibrated score more robust. Specifically, by using the standard deviation $\sigma_{\cdot|x_{<t}}$ as the calibration factor, we are essentially doing z-score normalization over $\log p(x_t|x_{<t})$, making the value of $\log p(x_t|x_{<t})$ more comparable across different cases. Later we will demonstrate that both factors contribute to the success of Min-K%++ through ablation study.

## 5 EXPERIMENTS

We conduct extensive experiments to validate the empirical efficacy of the proposed Min-K%++. We first focus on two established benchmarks for pre-training data detection. Following that, ablation studies are presented and discussed.

### 5.1 SETUP

**Benchmarks.** We focus on two benchmarks (and the only two to our knowledge) for pre-training data detection, WikiMIA (Shi et al., 2024) and MIMIR (Duan et al., 2024). WikiMIA is the first benchmark for pre-training data detection, which consists of texts from Wikipedia events. The training v.s. non-training data is determined by the timestamp. WikiMIA specifically groups data into splits according to the sentence length, intending to provide a fine-grained evaluation. It also considers two settings: *original* and *paraphrased*. The former assesses the detection of verbatim training texts, while the latter paraphrases the training texts (using ChatGPT) and evaluates on paraphrased inputs.

MIMIR (Duan et al., 2024) is built upon the Pile dataset (Gao et al., 2020), where training samples and non-training samples are drawn from the train and test split, respectively. MIMIR is found to be more challenging than WikiMIA since the training and non-training texts are from the same dataset and thus have minimal distribution shifts and temporal discrepancy (Duan et al., 2024).

**Baselines.** We consider 6 representative and state-of-the-art methods as our baselines, which are also featured in the benchmarking work of Duan et al. (2024). *Loss* method (Yeom et al., 2018) is a general technique that directly takes the loss as the score for detection. In the context of LLMs, this method is also reasonable as it is found that perplexity (the exponential of cross-entropy) can be a proxy for the occurrences of the training data (Gonen et al., 2023). Reference method (Carlini et al., 2021) (*Ref*) uses an extra LLM as reference to calibrate the likelihood of the input. *Zlib* and *Lowercase* method (Carlini et al., 2021) use zlib compression entropy and the likelihood of lowercased text as the reference to calibrate the likelihood, respectively. *Neighbor* method (Mattern et al., 2023) perturbs the input sentence with masked language models to create "neighbors" and calibrate the loss of the input sentence with the average loss of the neighbor sentences. Lastly, *Min-K%* (Shi et al., 2024) examines the exact token probability and averages a subset of minimum token scores over the input; it is currently the best-performing method on WikiMIA. For all methods, we either take the recommended configuration directly from the used benchmarks (Duan et al., 2024) or choose the hyperparameters with a hold-out validation set, following Shi et al. (2024). Later we will show that the proposed Min-K%++ is robust against its hyperparameter $k$ and obtains performance improvements over a wide range of $k$ values.

**Models.** WikiMIA is applicable to a wide range of models since Wikipedia dumps are often included into the training corpus of many LLMs. Specifically, we consider 5 families of models, including *Pythia* (Biderman et al., 2023) (2.8B, 6.9B, 12B), *GPT-NeoX* (Black et al., 2022) (20B), *LLaMA* (Touvron et al., 2023a) (13B, 30B, 65B), *OPT* (Zhang et al., 2022) (66B), and the new state-space model architecture *Mamba* (Gu & Dao, 2023) (1.4B, 2.8B). When a reference model is needed, following Shi et al. (2024) we use the smaller version correspondingly, *e.g.*, LLaMA-7B for LLaMA models and Pythia-70M for Pythia models. MIMIR is applicable to models that are trained on Pile. To be consistent with Duan et al. (2024), we focus on Pythia models (160M, 1.4B, 2.8B, 6.9B, 12B). In Appendix B, we provide further details to show that WikiMIA and MIMIR are indeed the right datasets for these models.

**Metrics.** As a binary classification problem, the detection performance can be evaluated with the AUROC score (area under the receiver operating characteristic curve) (Carlini et al., 2021; Shi et al., 2024; Duan et al., 2024). We define training data as "positive" and non-training data as "negative".

Table 1: AUROC results on WikiMIA benchmark (Shi et al., 2024). *Ori.* and *Para.* denote the original and paraphrased settings, respectively. **Bolded** number shows the best result within each column across all methods. The proposed Min-K%++ leads to remarkable improvements over existing methods in most settings.

| Len. | Method | Mamba-1.4B | | Pythia-6.9B | | LLaMA-13B | | LLaMA-30B | | LLaMA-65B | | Average | |
|---|---|---|---|---|---|---|---|---|---|---|---|---|---|
| | | *Ori.* | *Para.* | *Ori.* | *Para.* | *Ori.* | *Para.* | *Ori.* | *Para.* | *Ori.* | *Para.* | *Ori.* | *Para.* |
| 32 | Loss | 61.0 | 61.4 | 63.8 | 64.1 | 67.5 | 68.0 | 69.4 | 70.2 | 70.7 | 71.8 | 66.5 | 67.1 |
| | Ref | 62.2 | 62.3 | 63.6 | 63.5 | 57.9 | 56.2 | 63.5 | 62.4 | 68.8 | 68.2 | 63.2 | 62.5 |
| | Lowercase | 60.9 | 60.6 | 62.2 | 61.7 | 64.0 | 63.2 | 64.1 | 61.2 | 66.5 | 64.8 | 63.5 | 62.3 |
| | Zlib | 61.9 | 62.3 | 64.3 | 64.2 | 67.8 | 68.3 | 69.8 | 70.4 | 71.1 | 72.0 | 67.0 | 67.4 |
| | Neighbor | 64.1 | 63.6 | 65.8 | 65.5 | 65.8 | 65.0 | 67.6 | 66.3 | 69.6 | 68.7 | 66.6 | 65.8 |
| | Min-K% | 63.2 | 62.9 | 66.3 | 65.2 | 68.0 | 68.4 | 70.1 | 70.7 | 71.3 | 72.2 | 67.8 | 67.9 |
| | Min-K%++ | **66.8** | **66.1** | **70.3** | **68.0** | **84.8** | **82.7** | **84.3** | **81.2** | **85.1** | **81.4** | **78.3** | **75.9** |
| 64 | Loss | 58.2 | 56.4 | 60.7 | 59.3 | 63.6 | 63.1 | 66.2 | 65.5 | 67.9 | 67.7 | 63.3 | 62.4 |
| | Ref | 60.6 | 59.6 | 62.4 | 62.9 | 63.4 | 60.9 | 69.0 | 65.4 | 73.4 | 71.0 | 65.8 | 63.9 |
| | Lowercase | 57.0 | 57.0 | 58.2 | 57.7 | 62.0 | 61.0 | 62.1 | 59.8 | 64.5 | 61.9 | 60.8 | 59.5 |
| | Zlib | 60.4 | 59.1 | 62.6 | 61.6 | 65.3 | 65.3 | 67.5 | 67.4 | 69.1 | 69.3 | 65.0 | 64.5 |
| | Neighbor | 60.6 | 60.6 | 63.2 | 63.1 | 64.1 | 64.7 | 67.1 | 66.7 | 69.6 | 69.5 | 64.9 | 64.9 |
| | Min-K% | 62.2 | 58.0 | 65.0 | 61.1 | 66.0 | 64.0 | 68.5 | 65.7 | 69.8 | 67.9 | 66.3 | 63.3 |
| | Min-K%++ | **67.2** | **63.3** | **71.6** | **64.8** | **85.7** | **78.8** | **84.7** | **74.9** | **83.8** | **74.0** | **78.6** | **71.2** |
| 128 | Loss | 63.3 | 62.7 | 65.1 | 64.7 | 67.8 | 67.2 | 70.3 | 69.2 | 70.7 | 70.2 | 67.4 | 66.8 |
| | Ref | 62.0 | 61.1 | 63.3 | 62.9 | 62.6 | 59.7 | 71.9 | 70.0 | 73.7 | 72.0 | 66.7 | 65.1 |
| | Lowercase | 58.5 | 57.7 | 60.5 | 60.0 | 60.6 | 56.4 | 59.1 | 55.4 | 63.3 | 60.1 | 60.4 | 57.9 |
| | Zlib | 65.6 | 65.3 | 67.6 | **67.4** | 69.7 | 69.6 | 71.8 | 71.5 | 72.1 | **72.1** | 69.4 | 69.2 |
| | Neighbor | 64.8 | 62.6 | 67.5 | 64.3 | 68.3 | 64.0 | 72.2 | 67.2 | 73.7 | 70.3 | 69.3 | 65.7 |
| | Min-K% | 66.8 | 64.5 | 69.5 | 67.0 | 71.5 | 68.7 | 73.9 | 70.2 | 73.6 | 70.8 | 71.0 | 68.2 |
| | Min-K%++ | **68.8** | **65.6** | **70.7** | 66.8 | **83.9** | **76.2** | **82.6** | **73.8** | **80.0** | 70.7 | **77.2** | **70.6** |

AUROC is threshold-independent and can be interpreted as the the probability that a positive instance has higher score than a negative instance according to the detector. Therefore, the higher the better, and the random-guessing baseline is 50%. While we use AUROC as the main metric, we also report True Positive Rate (TPR) at low False Positive Rate (FPR) which measures detection rate at a meaningful threshold.

## 5.2 RESULTS

**WikiMIA results.** Table 1 shows major results in terms of AUROC; for results on more models and TPR numbers, please see Appendix D Tables 5 to 8. We remark that Min-K%++ achieves significant improvements over existing methods. In the original setting, Min-K%++ on average outperforms the runner-up Min-K% by {10.5%, 12.3%, 6.2%} with inputs of length {32, 64, 128}, respectively. In the paraphrased setting, Min-K%++ is also the best-performing approach on average in all cases.

**Min-K%++ is consistent across models and input lengths.** It can be noticed that Min-K%++'s superior results are agnostic to models: besides transformer-based LLMs, Min-K%++ also decently outperforms others on the new state space-based architecture, Mamba. In terms of input length, Shi et al. (2024) identify that short inputs are more challenging than longer inputs. While this is indeed the case for Min-K%, which yields {62.2%, 66.8%} AUROC on {64, 128}-length inputs with Mamba-1.4B (a 4.6% decrease when changing to shorter inputs), Min-K%++ achieves a much more consistent performances of {67.2%, 68.8%} (a mere decrease of 1.6%). Both observations demonstrate the robustness and generality of our proposed method.

**MIMIR results.** Table 2 shows the AUROC results; see Appendix D Table 9 for TPR results. Most numbers are taken from those reported by Duan et al. (2024). MIMIR is challenging in that the training and non-training texts are maximally similar to each other since they are drawn from the same dataset. Nonetheless, Min-K%++ still improves upon existing (reference-free) methods in most cases. Averaged over 7 subsets, Min-K%++'s relative AUROC w.r.t. the powerful Min-K% is {−0.2%, +0.5%, +1.1%, +1.8%, +2.8%} on Pythia model with {160M, 1.4B, 2.8B, 6.9B, 12B} parameters, respectively. Extrapolating this trend, we anticipate Min-K%++'s effectiveness to be even more obvious with larger models.

Table 2: AUROC results on the challenging MIMIR benchmark (Duan et al., 2024) with a suite of Pythia models (Biderman et al., 2023). In each column, the best result across all methods is **bolded**, with the runner-up underlined. †Ref relies on an extra reference LLM. ‡Neighbor induces significant extra computational cost than others ($25\times$ in this case), for which reason we don't run on the 12B model. Despite not requiring a reference model like the Ref method does, our Min-K%++ consistently achieves superior or comparable performance.

| | Wikipedia | | | | | Github | | | | | Pile CC | | | | | PubMed Central | | | | |
|---|---|---|---|---|---|---|---|---|---|---|---|---|---|---|---|---|---|---|---|---|
| Method | 160M | 1.4B | 2.8B | 6.9B | 12B | 160M | 1.4B | 2.8B | 6.9B | 12B | 160M | 1.4B | 2.8B | 6.9B | 12B | 160M | 1.4B | 2.8B | 6.9B | 12B |
| Loss | 50.2 | 51.3 | 51.8 | 52.8 | 53.5 | 65.7 | 69.8 | 71.3 | 73.0 | 74.0 | 49.6 | 50.0 | 50.1 | 50.7 | 51.1 | 49.9 | 49.8 | 49.9 | 50.6 | 51.3 |
| †Ref | 51.2 | 55.2 | 58.1 | 61.8 | 63.9 | 63.9 | 67.1 | 65.3 | 64.4 | 63.0 | 49.2 | 52.2 | 53.7 | 54.9 | 56.7 | 51.3 | 53.1 | 53.7 | 55.9 | 58.2 |
| Zlib | 51.1 | 52.0 | 52.4 | 53.5 | 54.3 | 67.4 | 71.0 | 72.3 | 73.9 | 74.8 | 49.6 | 50.1 | 50.3 | 50.8 | 51.1 | 49.9 | 50.0 | 50.1 | 50.6 | 51.2 |
| ‡Neighbor | 50.7 | 51.7 | 52.2 | 53.2 | / | 65.3 | 69.4 | 70.5 | 72.1 | / | 49.6 | 50.0 | 50.1 | 50.8 | / | 47.9 | 49.1 | 49.7 | 50.1 | / |
| Min-K% | 50.2 | 51.3 | 51.8 | 53.6 | 54.4 | 65.7 | 69.9 | 71.4 | 73.2 | 74.3 | 50.3 | 51.0 | 50.8 | 51.5 | 51.7 | 50.6 | 50.3 | 50.5 | 51.2 | 52.3 |
| Min-K%++ | 49.7 | 53.7 | 55.1 | 58.0 | 61.1 | 64.8 | 69.6 | 70.9 | 72.8 | 74.2 | 50.6 | 51.0 | 51.0 | 53.0 | 53.5 | 50.6 | 51.4 | 52.4 | 54.2 | 55.4 |

| | ArXiv | | | | | DM Mathematics | | | | | HackerNews | | | | | Average | | | | |
|---|---|---|---|---|---|---|---|---|---|---|---|---|---|---|---|---|---|---|---|---|
| Method | 160M | 1.4B | 2.8B | 6.9B | 12B | 160M | 1.4B | 2.8B | 6.9B | 12B | 160M | 1.4B | 2.8B | 6.9B | 12B | 160M | 1.4B | 2.8B | 6.9B | 12B |
| Loss | 51.0 | 51.5 | 51.9 | 52.9 | 53.4 | 48.8 | 48.5 | 48.4 | 48.5 | 48.5 | 49.4 | 50.5 | 51.3 | 52.1 | 52.8 | 52.1 | 53.1 | 53.5 | 54.4 | 54.9 |
| †Ref | 49.4 | 51.5 | 53.1 | 55.8 | 57.5 | 51.1 | 51.1 | 50.5 | 51.1 | 50.9 | 49.1 | 52.2 | 55.1 | 57.9 | 60.6 | 52.2 | 54.6 | 55.6 | 57.4 | 58.7 |
| Zlib | 50.1 | 50.9 | 51.3 | 52.2 | 52.7 | 48.1 | 48.2 | 48.0 | 48.1 | 48.1 | 49.7 | 50.3 | 50.8 | 51.2 | 51.7 | 52.3 | 53.2 | 53.6 | 54.3 | 54.8 |
| ‡Neighbor | 50.7 | 51.4 | 51.8 | 52.2 | / | 49.0 | 47.0 | 46.8 | 46.6 | / | 50.9 | 51.7 | 51.5 | 51.9 | / | 52.0 | 52.9 | 53.2 | 53.8 | / |
| Min-K% | 51.0 | 51.7 | 52.5 | 53.6 | 54.6 | 49.4 | 49.7 | 49.5 | 49.6 | 49.7 | 50.9 | 51.3 | 52.6 | 53.6 | 54.6 | 52.6 | 53.6 | 54.2 | 55.2 | 55.9 |
| Min-K%++ | 50.1 | 51.1 | 53.7 | 55.2 | 58.0 | 50.5 | 50.9 | 51.7 | 51.6 | 51.9 | 50.7 | 51.3 | 52.6 | 54.5 | 56.5 | 52.4 | 54.1 | 55.3 | 57.0 | 58.7 |

One powerful method on MIMIR is the Reference approach (Carlini et al., 2021) (Ref). However, it needs to perform inference with another LLM, which is expensive and much less feasible. Furthermore, the actual results of Ref on MIMIR are obtained by exhaustively trying out 8 different LLMs as the reference model and picking the best one (Duan et al., 2024). In contrast, our Min-K%++ does not rely on a reference model, yet provides competitive performance that is on par with Ref. Meanwhile, we remark that Min-K%++ achieves new SOTA results among all the 5 reference-free methods (as evidenced in the "Average" tab in Table 2. Lastly, in Appendix D Table 9, Min-K%++ outperforms all baselines when it comes to TPR performance.

**An online detection setting.** We further study an online detection setting to simulate a "detect-while-generating" scenario which can help minimize the generation of memorized and sensitive content. In a nutshell, we approximate such setting by adapting WikiMIA and perform experiments which have shown that the proposed method again outperform existing ones by noticeable margins. See Appendix C for details.

### 5.3 ABLATION STUDY

We focus on WikiMIA with LLaMA-13B model for ablation study.

**Ablation on the hyperparameter $k\%$.** $k$ determines what percent of token sequences with minimum scores are chosen to compute the final score. From Figure 4, it is obvious that Min-K%++ is robust to the choice of $k$, with the best and the worst result being 84.8% and 82.1% (a variation of 2.7%), respectively. Min-K% has a similar hyperparameter but is more sensitive to it: the variation between the best (68.0%) and the worst result (63.6%) is 4.4%, slightly larger than that of Min-K%++.

**Decomposing the contribution of calibration factors.** Recall that our method can be interpreted as calibrating the log probability $\log p(x_t|x_{<t})$ with two calibration factors $\mu_{\cdot|x_{<t}}$ and $\sigma_{\cdot|x_{<t}}$ (Equation (3)). In Table 3, we decompose the effect of $\mu_{\cdot|x_{<t}}$ and $\sigma_{\cdot|x_{<t}}$. Specifically, starting from the raw log probability $\log p(x_t|x_{<t})$, we gradually incorporate $\mu_{\cdot|x_{<t}}$ and $\sigma_{\cdot|x_{<t}}$ into the score computation, until we reach the final formulation of our Min-K%++. For example, when $\log p(x_t|x_{<t})$ and $\sigma_{\cdot|x_{<t}}$ are included (marked by ✓ in Table 3), the token-wise score becomes $\frac{\log p(x_t|x_{<t})}{\sigma_{\cdot|x_{<t}}}$. From the results, we see that calibrating the log probability with either $\mu_{x_{<t}}$ or $\sigma_{x_{<t}}$ alone already leads to 9.3% and 7.0% performance boosts over using raw log probability. Combining them together, which leads to the formulation specified by Min-K%++, takes advantage from both

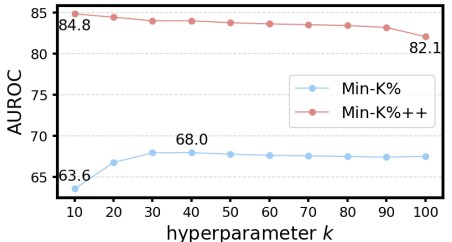

Figure 4: Ablation on $k$. Min-K%++ is robust to the hyperparameter choice.

Table 3: Decomposing the token-wise score of Min-K%++ in Equation (3). The last row corresponds to the final design of our Min-K%++.

| $\log p(x_t\|x_{<t})$ | $\mu_{x<t}$ | $\sigma_{x<t}$ | AUROC |
|:---:|:---:|:---:|:---:|
| ✓ | | | 68.0 |
| ✓ | ✓ | | 77.3 |
| ✓ | | ✓ | 75.0 |
| ✓ | ✓ | ✓ | 84.8 |

factors and results in a larger improvement of 16.8%. Such observation again validates the proposed method.

## 6 CONCLUSION AND DISCUSSION

In this work, we propose Min-K%++ as a novel method for pre-training data detection for LLMs. Motivated by our insight that training data tends to be local maximum or locates near local maximum along input dimensions, we design our method to examine whether the input forms the mode or has relatively high probability under the conditional categorical distribution of the LLM. It consistently achieves superior performances on two existing benchmarks and in various settings, which establishes a solid baseline for future studies. We hope that our method, along with our theoretical and empirical analysis, can motivate more research upon the pre-training data detection problem.

## ACKNOWLEDGEMENTS

This work is supported by NSF CNS-223380812, NSF CNS-2112562, and DoD W911NF-23-2-0224.

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

# A  COMPUTATIONAL COMPLEXITY

```python
# T: input token sequence length
# V: vocabulary size of the model
# input_ids [T,]: input token sequence
# logits [T, V]: raw outputs of the model
# probs [T, V]: softmax probability distribution over the vocabulary at
#                             each token position
# log_probs [T, V]: log softmax probabilities over the vocabulary at each
#                             token position
# token_log_probs [T,]: the log probability of each input token

#######################################################
# standard LLM inference; necessary for all methods
logits = model(input_ids, labels=input_ids)
probs = torch.nn.functional.softmax(logits[0, :-1], dim=-1)
log_probs = torch.nn.functional.log_softmax(logits[0, :-1], dim=-1)
token_log_probs = log_probs.gather(dim=-1, index=input_ids).squeeze(-1)

#######################################################
# operations specific to Min-K%++
# basic algebraic operations with negligible computational cost
mu = (probs * log_probs).sum(-1)
sigma_square = (probs * torch.square(log_probs)).sum(-1) - torch.square(
                                             mu)
# Equation (3)
mink_pp = (token_log_probs - mu) / sigma_square.sqrt()
# Equation (4)
mink_pp_aggregated = numpy.mean(np.sort(mink_pp)[:int(len(mink_pp * k))])
```

We show the python and pytorch-style pseudo-code above which implements Min-K%++. Concretely, the computational cost almost solely comes from the one-time LLM inference itself, which is shared by all methods. After obtaining the output statistics, the detection score of Min-K%++ are computed with basic algebraic operations, whose computational overhead is negligible.

In terms of comparison with baseline methods, Min-K%++ is as fast as the fastest ones (including Loss, Zlib, and Min-K%), as their execution all consists of one forward pass of the LLM and then some basic algebraic operations. Ref, Lowercase, and Neighbor all requires multiple forward passes of the LLM, thus being significantly slower than the other methods.

# B  DETAILS OF BENCHMARKS

Here we discuss more details of the used benchmarks and evaluated models to clarify their validity.

For WikiMIA, the training texts are from wiki articles before 2017, and the non-training texts are from wiki events that occurred post-2023 (Shi et al., 2024). Among the considered models, Pythia, GPT-NeoX, OPT, and Mamba models are all trained on the Pile dataset, which includes Wikipedia dumps in its corpora and was released to public in January 2021. The timestamps guarantee that non-training texts are not seen by the models. Similarly, for LLaMA models, they trained on Wikipedia dumps no later than August 2022.

MIMIR is constructed from the Pile dataset, where training v.s. non-training texts are directly determined by the train-test split provided by the Pile. Here we considered Pythia models which were trained on the training split of the Pile (and of course were not exposed to the test split), exactly following the evaluation setup in the work of MIMIR (Duan et al., 2024).

# C  AN ONLINE DETECTION SETTING

**Motivation.** Just like image generative models typically have a filter to screen harmful generated contents (Rombach et al., 2022), we believe building a similar mechanism for LLMs to detect memorized generated content in an online fashion would be helpful. Say, for example, that after

generating a few sentences, the model for some reason suddenly starts to emit copyrighted or private information that is memorized from the training data. In such case, an effective mitigation would be to use pre-training data detection method internally to identify that a part of the generated text is training data, and then let the model refrain from further generation.

**Setup.** Current benchmarks fail to simulate the online detection setting, since each whole input is either pure training text or pure non-training text. To address this, we adapt WikiMIA to construct an online version of the benchmark. Concretely, each input text is created by concatenating a training text at the end of a non-training text, closely simulating the representative scenario discussed above. Both the training and non-training text have random length, varying among $\{32, 64, 128\}$. In this online setting, the prediction on each part of the input, instead of on the whole input, is of interests. Therefore, we split each input into chunks with a length of 32. Methods will be operating on all chunks, and the performance is evaluated on the chunk level. Essentially, this setup simulates using a non-overlapping sliding window of size 32 to sweep over the texts and detecting whether each part within the sliding window is training data or not.

**Results.** Table 4 summarizes the results. Note that the Ref, Lowercase, and Neighbor method are no longer applicable or practical in online setting since they require repeated model inference. We see that Min-K%++ is still the most reliable method for online detection. The numbers are lower than those in the conventional offline setting (Table 1), which is expected because now we can only evaluate $\log p(x_t|x_{<t}, y)$ with $y$ being the prepended text, rather than the exact $\log p(x_t|x_{<t})$.

Table 4: AUROC results in the online detection setting with LLaMA models.

|          | 13B      | 30B      | 65B      |
|----------|----------|----------|----------|
| Loss     | 58.1     | 61.4     | 64.0     |
| Zlib     | 58.3     | 61.5     | 64.0     |
| Min-K%   | 58.4     | 61.6     | 64.1     |
| Min-K%++ | **68.1** | **67.4** | **68.4** |

## D   ADDITIONAL RESULTS

We present additional results here. Tables 5 and 6 show the AUROC results on WikiMIA with all the 10 models in the original and paraphrased setting, respectively. Tables 7 and 8 show the TPR at low FPR results. Table 9 shows the TPR at low FPR results on MIMIR.

Table 5: Full AUROC results on WikiMIA in the *original (verbatim training text)* setting.

| Len. | Model | Loss | Ref | Lowercase | Zlib | Neighbor | Min-K% | Min-K%++ |
|---|---|---|---|---|---|---|---|---|
| 32 | Mamba-1.4B | 61.0 | 62.2 | 60.9 | 61.9 | 64.1 | 63.2 | **66.8** |
| | Mamba-2.8B | 64.1 | 67.0 | 63.6 | 64.7 | 67.0 | 66.1 | **69.3** |
| | Pythia-2.8B | 61.4 | 61.3 | 60.9 | 62.1 | 64.2 | 61.8 | **64.4** |
| | Pythia-6.9B | 63.8 | 63.6 | 62.2 | 64.3 | 65.8 | 66.3 | **70.3** |
| | Pythia-12B | 65.4 | 65.1 | 64.8 | 65.8 | 66.6 | 68.1 | **72.3** |
| | NeoX-20B | 68.8 | 67.2 | 68.0 | 69.0 | 70.2 | 71.8 | **75.0** |
| | LLaMA-13B | 67.5 | 57.9 | 64.0 | 67.8 | 65.8 | 68.0 | **84.8** |
| | LLaMA-30B | 69.4 | 63.5 | 64.1 | 69.8 | 67.6 | 70.1 | **84.3** |
| | LLaMA-65B | 70.7 | 68.8 | 66.5 | 71.1 | 69.6 | 71.3 | **85.1** |
| | OPT-66B | 65.7 | 68.7 | 63.0 | 66.0 | 68.2 | 67.7 | **70.2** |
| | Average | 65.8 | 64.5 | 63.8 | 66.3 | 66.9 | 67.4 | **74.3** |
| 64 | Mamba-1.4B | 58.2 | 60.6 | 57.0 | 60.4 | 60.6 | 62.2 | **67.2** |
| | Mamba-2.8B | 61.2 | 64.3 | 61.7 | 63.0 | 63.6 | 65.4 | **70.6** |
| | Pythia-2.8B | 58.4 | 59.6 | 57.8 | 60.6 | 61.3 | 61.2 | **65.0** |
| | Pythia-6.9B | 60.7 | 62.4 | 58.2 | 62.6 | 63.2 | 65.0 | **71.6** |
| | Pythia-12B | 61.9 | 63.0 | 59.6 | 63.5 | 62.6 | 67.8 | **72.6** |
| | NeoX-20B | 66.2 | 65.7 | 65.8 | 67.6 | 67.1 | 72.2 | **76.0** |
| | LLaMA-13B | 63.6 | 63.4 | 62.0 | 65.3 | 64.1 | 66.0 | **85.7** |
| | LLaMA-30B | 66.2 | 69.0 | 62.1 | 67.5 | 67.1 | 68.5 | **84.7** |
| | LLaMA-65B | 67.9 | 73.4 | 64.5 | 69.1 | 69.6 | 69.8 | **83.8** |
| | OPT-66B | 62.3 | 67.0 | 61.2 | 63.9 | 64.1 | 67.0 | **70.0** |
| | Average | 62.7 | 64.8 | 61.0 | 64.4 | 64.3 | 66.5 | **74.7** |
| 128 | Mamba-1.4B | 63.3 | 62.0 | 58.5 | 65.6 | 64.8 | 66.8 | **68.8** |
| | Mamba-2.8B | 66.2 | 66.9 | 62.4 | 68.5 | 67.7 | 71.0 | **73.4** |
| | Pythia-2.8B | 62.8 | 59.6 | 59.5 | 65.0 | 65.2 | 66.8 | **66.8** |
| | Pythia-6.9B | 65.1 | 63.3 | 60.5 | 67.6 | 67.5 | 69.5 | **70.7** |
| | Pythia-12B | 65.8 | 63.9 | 61.4 | 67.8 | 67.1 | 70.7 | **72.7** |
| | NeoX-20B | 70.1 | 67.8 | 67.7 | 71.8 | 71.6 | 75.0 | **75.9** |
| | LLaMA-13B | 67.8 | 62.6 | 60.6 | 69.7 | 68.3 | 71.5 | **83.9** |
| | LLaMA-30B | 70.3 | 71.9 | 59.1 | 71.8 | 72.2 | 73.9 | **82.6** |
| | LLaMA-65B | 70.7 | 73.7 | 63.3 | 72.1 | 73.7 | 73.6 | **80.0** |
| | OPT-66B | 65.5 | 66.9 | 59.3 | 67.5 | 67.7 | 70.5 | **72.3** |
| | Average | 66.8 | 65.8 | 61.2 | 68.7 | 68.6 | 70.9 | **74.7** |

Table 6: Full AUROC results on WikiMIA in the *paraphrased* setting.

| Len. | Model | Loss | Ref | Lowercase | Zlib | Neighbor | Min-K% | Min-K%++ |
|---|---|---|---|---|---|---|---|---|
| 32 | Mamba-1.4B | 61.4 | 62.3 | 60.6 | 62.3 | 63.6 | 62.9 | **66.1** |
| | Mamba-2.8B | 64.5 | 66.6 | 63.5 | 64.8 | 66.3 | 65.3 | **67.9** |
| | Pythia-2.8B | 61.6 | 61.2 | 60.3 | 62.3 | **64.5** | 61.7 | 62.4 |
| | Pythia-6.9B | 64.1 | 63.5 | 61.7 | 64.2 | 65.5 | 65.2 | **68.0** |
| | Pythia-12B | 65.6 | 64.9 | 64.4 | 65.9 | 66.8 | 67.2 | **69.8** |
| | NeoX-20B | 68.2 | 66.3 | 66.7 | 68.2 | 68.3 | **69.7** | 69.6 |
| | LLaMA-13B | 68.0 | 56.2 | 63.2 | 68.3 | 65.0 | 68.4 | **82.7** |
| | LLaMA-30B | 70.2 | 62.4 | 61.2 | 70.4 | 66.3 | 70.7 | **81.2** |
| | LLaMA-65B | 71.8 | 68.2 | 64.8 | 72.0 | 68.7 | 72.2 | **81.4** |
| | OPT-66B | 65.3 | **68.2** | 62.7 | 65.4 | 66.7 | 66.3 | 68.1 |
| | Average | 66.1 | 64.0 | 62.9 | 66.4 | 66.2 | 67.0 | **71.7** |
| 64 | Mamba-1.4B | 56.4 | 59.6 | 57.0 | 59.1 | 60.6 | 58.0 | **63.3** |
| | Mamba-2.8B | 59.8 | 64.5 | 62.0 | 61.9 | 63.7 | 62.4 | **65.8** |
| | Pythia-2.8B | 56.5 | 59.2 | 56.1 | 59.0 | **59.6** | 56.8 | 58.5 |
| | Pythia-6.9B | 59.3 | 62.9 | 57.7 | 61.6 | 63.1 | 61.1 | **64.8** |
| | Pythia-12B | 60.0 | 63.2 | 59.1 | 62.1 | 62.8 | 62.5 | **65.8** |
| | NeoX-20B | 64.4 | 65.9 | 65.1 | 66.4 | 67.4 | 66.1 | **67.5** |
| | LLaMA-13B | 63.1 | 60.9 | 61.0 | 65.3 | 64.7 | 64.0 | **78.8** |
| | LLaMA-30B | 65.5 | 65.4 | 59.8 | 67.4 | 66.7 | 65.7 | **74.9** |
| | LLaMA-65B | 67.7 | 71.0 | 61.9 | 69.3 | 69.5 | 67.9 | **74.0** |
| | OPT-66B | 60.4 | **67.9** | 60.1 | 62.3 | 64.6 | 62.6 | 64.7 |
| | Average | 61.3 | 64.0 | 60.0 | 63.4 | 64.3 | 62.7 | **67.8** |
| 128 | Mamba-1.4B | 62.7 | 61.1 | 57.7 | 65.3 | 62.6 | 64.5 | **65.6** |
| | Mamba-2.8B | 65.7 | 66.6 | 61.2 | 68.3 | 64.6 | 68.0 | **70.0** |
| | Pythia-2.8B | 62.3 | 59.5 | 59.6 | **65.0** | 61.9 | 64.7 | 63.4 |
| | Pythia-6.9B | 64.7 | 62.9 | 60.0 | **67.4** | 64.3 | 67.0 | 66.8 |
| | Pythia-12B | 65.4 | 63.9 | 60.4 | 67.9 | 64.3 | 68.5 | **68.8** |
| | NeoX-20B | 69.5 | 67.8 | 67.4 | 71.8 | 69.6 | **72.6** | 72.2 |
| | LLaMA-13B | 67.2 | 59.7 | 56.4 | 69.6 | 64.0 | 68.7 | **76.2** |
| | LLaMA-30B | 69.2 | 70.0 | 55.4 | 71.5 | 67.2 | 70.2 | **73.8** |
| | LLaMA-65B | 70.2 | 72.0 | 60.1 | **72.1** | 70.3 | 70.8 | 70.7 |
| | OPT-66B | 64.5 | 66.8 | 57.4 | 66.9 | 63.4 | 67.2 | **68.2** |
| | Average | 66.1 | 65.0 | 59.5 | 68.6 | 65.2 | 68.2 | **69.6** |

Table 7: Full TPR at low FPR (FPR=5%) results on WikiMIA in the *original (verbatim training text)* setting.

| Len. | Model | Loss | Ref | Lowercase | Zlib | Neighbor | Min-K% | Min-K%++ |
|---|---|---|---|---|---|---|---|---|
| 32 | Mamba-1.4B | 14.2 | 7.8 | 11.1 | **15.5** | 11.9 | 14.7 | 12.9 |
| | Mamba-2.8B | 14.7 | 9.8 | 16.8 | 16.3 | 16.0 | **18.1** | 13.4 |
| | Pythia-2.8B | 14.7 | 6.2 | 11.1 | 15.8 | 15.0 | **17.1** | 14.2 |
| | Pythia-6.9B | 14.2 | 6.7 | 10.6 | 16.3 | 16.5 | **17.8** | 17.1 |
| | Pythia-12B | 17.1 | 9.0 | 16.3 | 17.1 | 19.4 | **23.0** | 18.6 |
| | NeoX-20B | 19.9 | 15.5 | 18.1 | 19.9 | 22.2 | **27.9** | 19.4 |
| | LLaMA-13B | 13.9 | 4.7 | 9.6 | 11.6 | 11.6 | 18.9 | **38.5** |
| | LLaMA-30B | 18.4 | 9.8 | 11.4 | 14.5 | 9.3 | 21.2 | **31.3** |
| | LLaMA-65B | 22.5 | 12.4 | 12.1 | 18.6 | 6.5 | 26.1 | **41.1** |
| | OPT-66B | 14.2 | 10.8 | 10.6 | 16.0 | 21.7 | **22.0** | 19.4 |
| | Average | 16.4 | 9.3 | 12.8 | 16.1 | 15.0 | 20.7 | **22.6** |
| 64 | Mamba-1.4B | 9.5 | 4.6 | 8.8 | 14.1 | 8.8 | **19.4** | 16.6 |
| | Mamba-2.8B | 10.2 | 9.2 | 16.6 | 14.8 | 10.6 | 19.0 | **21.5** |
| | Pythia-2.8B | 10.2 | 10.6 | 10.2 | 14.4 | 10.2 | **18.3** | 16.2 |
| | Pythia-6.9B | 13.4 | 12.0 | 11.6 | 16.2 | 10.9 | 19.0 | **26.1** |
| | Pythia-12B | 9.2 | 13.0 | 12.3 | 11.3 | 11.3 | **21.5** | 20.1 |
| | NeoX-20B | 13.0 | 15.5 | 15.5 | 16.6 | 13.0 | **20.4** | 20.4 |
| | LLaMA-13B | 11.3 | 4.2 | 11.6 | 12.7 | 10.2 | 17.2 | **34.1** |
| | LLaMA-30B | 13.7 | 11.3 | 11.3 | 15.5 | 9.9 | 17.6 | **36.3** |
| | LLaMA-65B | 15.1 | 13.0 | 12.3 | 16.9 | 9.9 | 18.0 | **38.4** |
| | OPT-66B | 13.4 | 13.0 | 10.9 | 13.4 | 12.0 | **26.4** | 22.5 |
| | Average | 11.9 | 10.6 | 12.1 | 14.6 | 10.7 | 19.7 | **25.2** |
| 128 | Mamba-1.4B | 11.5 | 10.1 | 12.9 | **19.4** | 15.8 | 16.6 | 16.6 |
| | Mamba-2.8B | 19.4 | 10.1 | 13.7 | 23.7 | 15.1 | **25.9** | 21.6 |
| | Pythia-2.8B | 9.3 | 10.1 | 10.8 | **18.7** | 8.6 | 13.7 | 17.3 |
| | Pythia-6.9B | 14.4 | 13.7 | 12.9 | 20.9 | 10.8 | 18.0 | **22.3** |
| | Pythia-12B | 18.0 | 12.2 | 12.9 | 23.7 | 10.1 | **25.2** | 20.9 |
| | NeoX-20B | 18.7 | 15.8 | 12.2 | 23.0 | 15.8 | **25.2** | 23.0 |
| | LLaMA-13B | 21.6 | 10.8 | 15.8 | 18.7 | 12.9 | 25.9 | **43.2** |
| | LLaMA-30B | 23.7 | 10.8 | 10.1 | 18.0 | 15.1 | 23.7 | **40.3** |
| | LLaMA-65B | 23.0 | 18.0 | 14.4 | 22.3 | 15.8 | 23.7 | **27.3** |
| | OPT-66B | 20.9 | 17.3 | 14.4 | 21.6 | 12.9 | **23.0** | 16.6 |
| | Average | 18.1 | 12.9 | 13.0 | 21.0 | 13.3 | 22.1 | **24.9** |

Table 8: Full TPR at low FPR (FPR=5%) results on WikiMIA in the *paraphrased* setting.

| Len. | Model | Loss | Ref | Lowercase | Zlib | Neighbor | Min-K% | Min-K%++ |
|---|---|---|---|---|---|---|---|---|
| 32 | Mamba-1.4B | 14.2 | 5.9 | 13.2 | 13.2 | 7.2 | **15.2** | 10.6 |
| | Mamba-2.8B | 16.5 | 10.1 | 15.0 | 12.7 | 9.3 | **19.9** | 13.4 |
| | Pythia-2.8B | 14.2 | 7.2 | 11.6 | 14.5 | 8.5 | **16.5** | 13.9 |
| | Pythia-6.9B | 15.0 | 6.2 | 11.9 | 12.7 | 9.6 | **21.7** | 17.1 |
| | Pythia-12B | 17.3 | 8.0 | 16.5 | 15.5 | 9.8 | **19.9** | 17.3 |
| | NeoX-20B | 18.1 | 15.2 | 15.5 | 18.6 | 15.2 | **19.6** | 12.9 |
| | LLaMA-13B | 16.3 | 5.4 | 9.6 | 15.0 | 8.5 | 17.6 | **35.9** |
| | LLaMA-30B | 14.7 | 7.5 | 12.7 | 15.0 | 9.3 | 18.1 | **27.4** |
| | LLaMA-65B | 23.3 | 9.3 | 11.9 | 16.5 | 12.1 | 24.3 | **35.9** |
| | OPT-66B | 15.2 | 10.3 | 13.4 | 17.1 | 12.1 | **18.1** | 15.2 |
| | Average | 16.5 | 8.5 | 13.1 | 15.1 | 10.2 | 19.1 | **20.0** |
| 64 | Mamba-1.4B | 8.1 | 8.1 | 9.5 | **15.1** | 9.5 | 8.4 | 7.0 |
| | Mamba-2.8B | 12.3 | 11.3 | 14.8 | 14.8 | **18.3** | 13.0 | 12.3 |
| | Pythia-2.8B | 9.5 | 13.0 | 11.3 | **16.6** | 11.3 | 11.3 | 9.9 |
| | Pythia-6.9B | 10.6 | **16.2** | 11.3 | 15.8 | 12.7 | 12.7 | 14.1 |
| | Pythia-12B | 11.6 | 14.4 | 13.4 | **16.2** | 10.6 | 14.4 | 13.7 |
| | NeoX-20B | 16.2 | 14.1 | 13.7 | **19.4** | 18.3 | 17.6 | 13.0 |
| | LLaMA-13B | 12.0 | 4.6 | 13.7 | 13.4 | 14.4 | 13.4 | **26.4** |
| | LLaMA-30B | 13.4 | 8.1 | 8.1 | 16.9 | 11.6 | 14.4 | **21.5** |
| | LLaMA-65B | 13.4 | 10.9 | 9.5 | 18.0 | 16.9 | 13.7 | **29.2** |
| | OPT-66B | 13.4 | 13.0 | 13.4 | **14.8** | 13.7 | 14.8 | 12.7 |
| | Average | 12.0 | 11.4 | 11.9 | **16.1** | 13.7 | 13.4 | 16.0 |
| 128 | Mamba-1.4B | 13.7 | 11.5 | 11.5 | **17.3** | 13.7 | 14.4 | 10.1 |
| | Mamba-2.8B | 16.6 | 10.8 | 15.1 | **20.1** | 17.3 | 20.1 | 15.1 |
| | Pythia-2.8B | 14.4 | 7.2 | 8.6 | **16.6** | 12.2 | 14.4 | 14.4 |
| | Pythia-6.9B | 16.6 | 8.6 | 11.5 | 20.9 | 17.3 | 17.3 | **21.6** |
| | Pythia-12B | 19.4 | 8.6 | 12.2 | 19.4 | 10.1 | **21.6** | 17.3 |
| | NeoX-20B | 15.8 | 19.4 | 16.6 | 21.6 | 18.7 | **22.3** | 19.4 |
| | LLaMA-13B | 18.0 | 4.3 | 15.8 | 21.6 | 13.7 | 20.1 | **35.2** |
| | LLaMA-30B | 18.7 | 18.7 | 13.7 | 19.4 | 14.4 | 18.7 | **21.6** |
| | LLaMA-65B | 24.5 | 12.9 | 13.7 | 22.3 | 18.7 | **25.2** | 25.2 |
| | OPT-66B | 18.0 | 15.8 | 11.5 | 18.7 | 12.9 | **20.1** | 18.7 |
| | Average | 17.6 | 11.8 | 13.0 | 19.8 | 14.9 | 19.4 | **19.9** |

Table 9: TPR at low FPR (FPR=5%) results on MIMIR. In each column, the best result across all methods is **bolded**, with the runner-up underlined. [†]Ref relies on an extra reference LLM. [‡]Neighbor induces significant extra computational cost than others ($25\times$ in this case), for which reason we don't run on the 12B model.

| Method | Wikipedia | | | | | Github | | | | | Pile CC | | | | | PubMed Central | | | | |
|---|---|---|---|---|---|---|---|---|---|---|---|---|---|---|---|---|---|---|---|---|
| | 160M | 1.4B | 2.8B | 6.9B | 12B | 160M | 1.4B | 2.8B | 6.9B | 12B | 160M | 1.4B | 2.8B | 6.9B | 12B | 160M | 1.4B | 2.8B | 6.9B | 12B |
| Loss | 4.2 | 4.7 | 4.7 | 5.1 | 5.0 | 22.6 | 32.1 | 33.6 | 38.5 | 40.7 | 3.1 | 5.0 | 4.8 | 4.9 | 5.1 | 4.0 | 4.4 | 4.3 | 4.9 | 5.0 |
| [†]Ref | 6.1 | 5.3 | 5.5 | 5.6 | 5.7 | 23.4 | 14.8 | 14.9 | 15.4 | 16.2 | 5.5 | 5.6 | **5.8** | 5.8 | **7.5** | **5.7** | 4.1 | 4.0 | 5.9 | 8.7 |
| Zlib | 4.2 | 5.7 | 5.9 | 6.3 | 6.8 | 25.0 | 32.8 | **36.1** | **39.3** | 40.8 | 4.0 | 5.1 | 5.4 | 6.2 | 6.6 | 3.8 | 3.6 | 3.5 | 4.3 | 4.4 |
| [‡]Neighbor | 4.0 | 4.5 | 4.9 | 5.8 | / | 24.7 | 31.6 | 29.8 | 34.1 | / | 3.9 | 3.6 | 4.0 | 5.3 | / | 3.9 | 3.7 | 4.5 | 4.5 | / |
| Min-K% | **6.4** | 5.6 | 6.4 | 6.5 | **8.1** | 23.3 | 32.2 | 34.0 | 39.0 | **40.8** | 4.2 | 5.1 | 5.2 | 5.5 | 5.7 | 4.7 | 5.2 | 4.8 | 5.9 | 5.4 |
| Min-K%++ | 5.7 | **6.1** | **8.5** | **11.4** | **11.5** | **25.4** | **33.2** | 34.2 | 38.2 | 40.1 | **5.8** | 5.0 | 5.4 | **6.3** | 6.3 | 5.1 | **6.3** | **6.5** | **7.4** | **9.0** |

| Method | ArXiv | | | | | DM Mathematics | | | | | HackerNews | | | | | Average | | | | |
|---|---|---|---|---|---|---|---|---|---|---|---|---|---|---|---|---|---|---|---|---|
| | 160M | 1.4B | 2.8B | 6.9B | 12B | 160M | 1.4B | 2.8B | 6.9B | 12B | 160M | 1.4B | 2.8B | 6.9B | 12B | 160M | 1.4B | 2.8B | 6.9B | 12B |
| Loss | 4.0 | 4.8 | 4.6 | 5.4 | 5.6 | 3.8 | 4.3 | 4.1 | 4.1 | 4.0 | 5.0 | 4.8 | 5.5 | 5.9 | 6.8 | 6.7 | 8.6 | 8.8 | 9.8 | 10.3 |
| [†]Ref | 5.2 | 5.4 | 5.9 | 6.8 | 7.4 | 5.3 | 3.6 | 4.9 | 5.4 | 5.9 | 5.2 | **6.3** | **7.6** | **7.0** | **7.6** | 8.1 | 6.4 | 6.9 | 7.4 | 8.4 |
| Zlib | 2.9 | 4.3 | 4.1 | 4.6 | 4.7 | 4.1 | 5.0 | 4.6 | 4.3 | 4.3 | 5.0 | 5.5 | 5.8 | 5.6 | 5.8 | 7.0 | 8.9 | 9.3 | 10.1 | 10.5 |
| [‡]Neighbor | 4.7 | 4.8 | 4.4 | 4.1 | / | **5.6** | 4.4 | 4.5 | 4.5 | / | **6.5** | 5.2 | 5.3 | 5.7 | / | 7.6 | 8.3 | 8.2 | 9.1 | / |
| Min-K% | 4.9 | 4.8 | 4.7 | 5.6 | 6.2 | 4.5 | 4.5 | 4.6 | 4.7 | 5.2 | 5.2 | 5.7 | 5.9 | 6.3 | 6.9 | 7.6 | 9.0 | 9.4 | 10.5 | 11.2 |
| Min-K%++ | **6.0** | **6.0** | **6.7** | **8.2** | **8.6** | 5.4 | **5.5** | **5.7** | **6.2** | **6.3** | 5.5 | 4.9 | 5.7 | 6.6 | 6.6 | **8.4** | **9.6** | **10.4** | **12.0** | **12.6** |

