# OpenReview forum: "Min-K%++: Improved Baseline for Pre-Training Data Detection from Large Language Models"
_ICLR.cc/2025/Conference — ICLR 2025 Spotlight_

### Official Review · Reviewer_o61x · 2024-10-18

**Soundness:** 4
**Presentation:** 4
**Contribution:** 4
**Rating:** 8
**Confidence:** 5

**Summary:**

This paper presents a novel pretraining data detection method, theoretically identified as the optimal solution. It reframes the data detection problem as one of identifying local maxima, utilizing average and standard deviation metrics rather than relying on reference models. Experiments across two benchmarks and five models highlight the effectiveness of MinK++.

**Strengths:**

1. The motivation is clear and makes sense, which has been demonstrated by theoretical analysis.
2. The method is very simple and effective, which impresses me.
3. The experiments are very solid across two benchmarks, 5 models, and 6 baselines, which also highlights the effectiveness of MinK++.
4. The writing is good and easy to follow.

**Weaknesses:**

The work is very solid and has no more significant weaknesses.

1. It would be better to add some impactful works of 2024 in related works, including `A Careful Examination of Large Language Model Performance on Grade School Arithmetic` [1] and `Benchmarking Benchmark Leakage in Large Language Models` [2].

[1] A Careful Examination of Large Language Model Performance on Grade School Arithmetic. https://arxiv.org/abs/2405.00332

[2] Benchmarking Benchmark Leakage in Large Language Models. https://arxiv.org/abs/2404.18824

**Questions:**

I came across the DC-PDD paper [1], which concluded that MinK++ performs worse than MinK on their PatentMIA Benchmark. I'm curious to understand the reasons behind this. If you have the time and are willing to share your thoughts, I would greatly appreciate it. Of course, I understand if you're too busy and can't respond.

[1] Pretraining Data Detection for Large Language Models: A Divergence-based Calibration Method. https://arxiv.org/pdf/2409.14781

---

> ### Author Response · Authors · 2024-11-20
>
> We sincerely thank the reviewer for acknowledging our work and giving us constructive and insightful feedbacks. Below we address your comments/questions.
>
> ------------
>
>
> **W1: Related works**
>
>
> Thank you for pointing out these impactful works, which helps make the related work section more comprehensive. We have now added discussions about them in Section 2 in the revised manuscript (marked in blue).
>
>
> **Q1: Discussion on PatentMIA**
>
>
> Thank you for this thought-provoking question. Recall that by connecting with the score matching objective, we have shown that training data tends to form local maxima, which is the foundation of our proposed method. However, in practice, we believe that how well the score matching objective is really optimized on a certain input may involve multiple intertwined factors (e.g., model size, training steps, sample frequency, etc). When the objective is underoptimized on certain inputs, the Min-K%++ score may not be as informative as for other cases. For example, we hypothesize that texts from PatentMIA may contribute a quite small portion of the whole training corpus, thus causing the training optimization on them less complete, potentially explaining the slightly worse performance of Min-K%\++. Further understanding and quantifying the relationship between optimization and Min-K%++ score will be our future direction.

---

> > ### Comment · Reviewer_o61x · 2024-11-21
> > **Thank you for your reply**
> >
> > Thank you for your detailed reply and answer, which makes sense as the texts from PatentMIA are definitely rare in the training data. I will maintain my score.

---

> > > ### Author Response · Authors · 2024-11-26
> > > **Thank you**
> > >
> > > We sincerely thank the reviewer again for reviewing our work and giving us positive feedback. We greatly appreciate your insights and suggestions, which have helped improve our work in various aspects.

---

### Official Review · Reviewer_uqhR · 2024-11-03

**Soundness:** 3
**Presentation:** 3
**Contribution:** 3
**Rating:** 8
**Confidence:** 3

**Summary:**

This paper proposes a methodology for pre-training data detection, Min-K%++. They reduce the problem as identification of local maxima. Their method achieves state of the art results across multiple settings.

**Strengths:**

The paper is well written and it is easy to follow.

The paper shows the best results (or second best when reference models are used for the MIMIR benchmark) reported on the existing benchmarks, and compares with 6 established baselines for MIA task.

Figure 2 and Figure 3 in the paper provide a very good  summary of  the idea that the paper follows (first the concept of local maxima for identification of training examples, and second how it contrasts with prior work that solely looks at token probability).

**Weaknesses:**

Line 323: any additional benchmarks you could think of designing for this task? there are new open source LLMs for which the training data is also provided. I think the paper could be stronger if it is not only relying on existing benchmarks for MIA task, and devises its own.

I think the paper could improve if you clarify why MIMIR and WIKIMIA are the right datasets to measure this for the models used for evaluation. Line 347 and 352 touch on it but do not expand on why the current models have not seen the entire set. I am more curious about WIKIMIA for recent models, and how they are comparable as they are trained at different times.

**Questions:**

Line 147. Is Wang et al . 2022 the right ppaer to cite to refer to auto-regressive LLM maximum log likelihood training?

I wonder if the findings in this paper may merit a discussion and a mention in comparison to what is presented in yours. https://arxiv.org/pdf/2212.04037, thoughts?

Line 328: could you elaborate how the paraphrased inputs are created for the WIKIMIA dataset? How is it ensured that the paraphrases are accurate?

Line 460: why focus on WIKIMIA and Llama-13B ?

---

> ### Author Response · Authors · 2024-11-20
>
> We gratefully thank the reviewer for giving us constructive feedbacks and acknowledging our good empirical results plus clear presentation of our ideas. Below we address your comments/questions.
>
> -------------
>
> **W1: Any additional benchmarks?**
>
> At the time when we were conducting this work, there were only two benchmarks available for this task (WikiMIA and MIMIR), likely due to that this task itself is quite new (and we are among the very first to develop a methodology for it). Here we evaluate on a later released benchmark from Meeus et al. [1] which contains book texts. The benchmark is applicable to OpenLLaMA models [2], and below we show the detection AUROC on OpenLLaMA-7B and OpenLLaMA-13B. We see that Min-K%\++ still achieves superior results over competitive baselines.
>
> |           | OpenLLaMA-7B | OpenLLaMA-13B |
> |-----------|:------------:|:-------------:|
> | Loss      |     54.6     |      54.9     |
> | Ref       |     55.6     |      53.3     |
> | Lowercase |     54.7     |      53.3     |
> | Zlib      |     54.7     |      55.0     |
> | Neighbor  |     55.3     |      55.5     |
> | Min-K%    |     56.9     |      56.7     |
> | Min-K%++  |     **60.0**     |      **60.1**     |
>
> [1] Did the Neurons Read your Book? Document-level Membership Inference for Large Language Models
>
> [2] OpenLLaMA: An Open Reproduction of LLaMA
>
> **W2: Why MIMIR and WikiMIA are the right datasets that evaluated models have not seen the entire set?**
>
> - For WikiMIA, the training texts are from wiki articles before 2017, and the non-training texts are from wiki events that occurred post-2023. Among the considered models, Pythia, GPT-NeoX, OPT, and Mamba models are all trained on the Pile dataset, which includes Wikipedia dumps in its corpora and was released to public in January 2021. The timestamps guarantee that non-training texts are not seen by the models. Similarly, for LLaMA models, they trained on Wikipedia dumps no later than August 2022 (all information is from each corresponding paper). Also note that we closely follow the setup in the work of WikiMIA (Shi et al. at ICLR'24).
>
> - MIMIR is constructed from the Pile dataset, where training v.s. non-training texts are directly determined by the train-test split provided by the Pile. Here we considered Pythia models which were trained on the training split of the Pile (and of course were not exposed to the test split), exactly following the evaluation setup in the work of MIMIR (Duan et al. at COLM'24).
>
>
> We have now included these details in Appendix B and referenced them in Section 5.1 (marked in blue).
>
>
> **Q1: Reference for maximum likelihood training**
>
> Thank you for pointing this out. Indeed, the work of Wang et al. might not be the most approriate reference for this. We have now cited the work of GPT-2 and GPT-3 instead, where the advent of GPT-2 started the trend of using autoregressive maximum likelihood-based objective, according to Wang et al.
>
>
> **Q2: Discussion of a related work**
>
> One finding of the mentioned work (Gonen et al.), which is particular relevant to the training data detection problem, is that the perplexity can be a proxy for the occurrences of training data. This indeed to some extent relates to our finding that training data forms local maxima in terms of likelihood. In addition, the finding of Gonen et al. can be seen as a foundation for using the loss value to detect training data (perplexity is the exponential of cross-entropy loss), which is a baseline method we compare with. We have cited Gonen et al. and added relevant discussion in the revised manuscript in Section 5.1 "Baselines".
>
> -----------------------
> [1/2] of authors' response

---

> ### Author Response · Authors · 2024-11-20
>
> **Q3: Paraphrased inputs**
>
> We'd like to first note that although WikiMIA proposed to evaluate in paraphrased setting, the authors neither disclosed any detail on how the paraphrasing is done, nor released the paraphrased version of WikiMIA. Therefore, we need to do the paraphrasing ourselves in a way that we believe is reasonable.
>
>
> Specifically, we paraphrase inputs using ChatGPT and we try our best to ensure the validity with two steps. First, we instruct ChatGPT to paraphrase input sentence by substituting a certain number of words (around 5% of the sentence length) without changing anything else. We consider word replacement rather than extensive paraphrasing to avoid any potential ambiguity: Whether extensively paraphrased input can be determined exactly as the memorization of a certain sentence could be subjective. Then, we manually inspect each paraphrased input to make sure that it is clearly a near-verbatim of the original one, again to best avoid ambiguity. An example is shown below:
> - original: "The 12th Circle Chart Music Awards ceremony was held at KSPO Dome in Seoul on February 18, 2023, to recognise the best artists and recordings, primarily based on Circle Music Chart of..."
> - paraphrased: "The 12th Circle Chart Music Awards ceremony took place at KSPO Dome in Seoul on February 18, 2023, to celebrate the finest artists and recordings, chiefly based on Circle Music Chart of"
>
> Here, we can see that all key information is kept the same, and the paraphrased input is obviously a near-duplicate of the original one, meaning that indeed we would want to detect it as a memorized training data.
>
>
> **Q4: Why WikiMIA and LLaMA-13B in ablation study**
>
> There isn't any particular reason; we just focus on WikiMIA and LLaMA-13B as a representative setting. All observations hold across benchmarks and models.
>
>
> For example, on WikiMIA but with the Mamba-2.8B model, ablating the calibration factors (Table 3) we have the AUROC of 66.1 (without $\mu$ or $\sigma$), 68.0 (with $\mu$), 66.3 (with $\sigma$), and 69.3 (with both $\mu$ and $\sigma$). Varying the hyperparameter $k$ (Figure 4) we have the best and worst AUROC being 69.3 and 67.0, respectively (whereas the best AUROC of Min-K% is 65.4). Similarly, on MIMIR (DM Mathematics) and with Pythia-6.9B model, ablating the calibration factors (Table 3) we have the AUROC of 49.5 (without $\mu$ or $\sigma$), 51.1 (with $\mu$), 50.6 (with $\sigma$), and 51.7 (with both $\mu$ and $\sigma$). Varying the hyperparameter $k$ (Figure 4) we have the best and worst AUROC being 51.6 and 50.5, respectively (whereas the best AUROC of Min-K% is 49.6). These results all echo our observations that 1) both calibration factors contribute to the performance improvement, and 2) Min-K%++ is robust to hyperparameter.
>
> ----------------
> [2/2] of authors' response

---

> ### Comment · Reviewer_uqhR · 2024-11-21
> **Thanks for the response.**
>
> Thanks for the response and clarifications, and for running experiment on the additional datasets. I increased my score based on the above discussion.

---

> > ### Author Response · Authors · 2024-11-26
> > **Thank you**
> >
> > We sincerely thank the reviewer again for reviewing our work and giving us positive feedback. We greatly appreciate your insights and suggestions, which have helped improve our work in various aspects.

---

### Official Review · Reviewer_5hJP · 2024-11-03

**Soundness:** 3
**Presentation:** 3
**Contribution:** 2
**Rating:** 6
**Confidence:** 4

**Summary:**

This paper provides a MIA (Membership Inference Attack) method to detect whether data was used to train models, building upon previous MIN-K methods. Instead of using absolute logit likelihood values, they propose subtracting the mean of previous token likelihoods to identify how far the current value deviates from the previous distribution. This method shows improvements over MIN-K on two benchmarks, WIKIMIA and MIMIR, across different models.

**Strengths:**

- The research topic is important and worth more work for it.
- I love the motivation based on ISM objects, but sadly it seems that this does not lead to a well-established method based on this.

**Weaknesses:**

- The theoretical connection between Section 4.1's gradient conditions ($\frac{\partial \log p(x)}{\partial x_i} = 0$ and $\frac{\partial^2 \log p(x)}{\partial x_i^2} < 0$) and the proposed normalized score method ($\frac{\log p(x_t|x_{<t}) - \mu_{x_{<t}}}{\sigma_{x_{<t}}}$) for min-k% token selection needs clearer explanation. The current presentation shows a gap between the theoretical motivation and practical implementation.

- The justification in lines 256-263 regarding the method's effectiveness on rare or difficult-to-learn training texts requires stronger argumentation. For rare texts, distinguishing between untrained and rare samples remains challenging regardless of whether using direct scores or mean-subtracted scores. The same applies to inherently difficult-to-learn texts. This limitation needs to be addressed more explicitly.

- The state-of-the-art landscape has evolved beyond Min-K. Recent methods like [ReCall](https://arxiv.org/pdf/2406.15968) have demonstrated significant improvements, achieving nearly 20% performance gains over Min-K on the WIKIMIA benchmark. Including comparisons with these newer methods would strengthen the paper's relevance. While Min-K was among the first to apply MIA for data contamination detection, its limitations based on its underlying assumptions should be acknowledged.

- The MIMIR benchmark results, where most methods show minimal performance differences, warrant further discussion. The assumption that reference-based methods should naturally outperform reference-free approaches needs reconsideration, as these references differ from ground-truth evaluations. Therefore, achieving comparable performance to reference-based methods could serve as compelling evidence for method validation.

**Questions:**

I think conceptually we should have several groups of data to be classified, and these groups should be 'imagined' by researchers who are truly spending time looking at the *pattern of pre-training corpora*. Given a pre-trained chunk $s = \langle s_1,s_2,...,s_n\rangle$ with n tokens, we need to consider the false positive and false negative rates of proposed methods to identify:

1. Subportions of text $s' = \langle s_i,...,s_j\rangle$
2. Rare sequences versus untrained sequences
3. Rare tokens $s_i$ within a sequence

I am concerned that there are many nuances here, and these methods seem to capture them not very effectively.

---

> ### Author Response · Authors · 2024-11-20
>
> We thank the reviewer for giving us constructive feedbacks. Below we address your comments/questions.
>
>
> Meanwhile, we'd like to clarify in your summary that our method is not subtracting the mean of "previous token likelihoods". Rather, $\mu_{x_{<t}}$ is the mean of token likelihoods over the vocabulary when predicting the current token (conditioned on previous tokens $x_{<t}$). Essentially, we are comparing two likelihoods under the same conditional distribution which is $p(\cdot|x_{<t})$. The first is the likelihood of the target token, $\log p(x_t|x_{<t})$, and the second is the expected likelihood of all candidate tokens, $ \mu_{x_{<t}}=\mathbb{E}_{z \sim p(\cdot| x _{<t}) } [\log p(z|x _{<t})] $.
>
>
> -----
>
>
> **W1: Gap between theoretical motivation and practical implementation**
>
>
> Thank you for raising this point, which helped us refine our thinking. We have updated the relevant part in Section 4.2 with the discussions below to improve clarity.
>
>
> Recall our theoretical insight of "training data forms or locates near a local maximum (along each input dimension)", meaning that the training input tends to have higher probability than other neighbor input values. Under the discrete categorical distribution of LLMs, the insight can anologously translate to that the training token tends to have higher probability relative to many other candidate tokens in the vocabulary. Our formulation naturally reflects this idea by comparing $\log p(x_t|x_{<t})$ (probability of the input token) with $\mu_{x<t}$ (representative for the probability of other candidate tokens).
>
>
>
> **W2: Rare and difficult-to-learn texts**
>
>
> Indeed, detecting training texts that are rare and difficult-to-learn could be challenging for all methods, and we did not state (or intended to do so) that our method has completely solved this problem. We have now added discussion in Section 4.2 Interpretation 1 to make this more clear.
>
>
> Yet, we believe that our method is more effective than existing methods in addressing this challenge, given the concrete example discussed with Figure 3. Specifically, when the training text is rare or difficult-to-learning, the predicted distribution across the vocabulary could be, say, near uniform, rendering the probability of the training token $\log p(x_t|x_{<t})$ being low. As a result, existing methods like Min-K% may mark it as non-training since it looks at the direct likelihood. In contrast, our method measures the relative relationship $\log p(x_t|x_{<t})-\mu_{x_{<t}}$, which helps detect training data despite the low absolute value of $\log p(x_t|x_{<t})$ itself.
>
>
>
> **W3: Latest works beyond Min-K%**
>
>
> Indeed, as the attention on this problem rapidly grows, we ourselves also see more works coming out recently. However, we clarify that at the time when we conducted our work, Min-K% was still one of the few works that targets LLM training data detection and was the state-of-the-art. In fact, our work is among the first that improve upon Min-K% and is prior to many recent works, including the mentioned arXiv preprint ReCall (we do have evidence, but to preserve the anonymity we are unable to disclose it). Due to this reason, we believe Min-K% serves as a strong and valid baseline. Comparing with Min-K% thus effectively demonstrates the efficacy and significance of our method.
>
>
> ------------------
> [1/2] of authors' response

---

> ### Author Response · Authors · 2024-11-20
>
> **W4: The validity of reference-based method being a competitive baseline**
>
>
> The reference-based method (Carlini et al.; denoted as Ref in the manuscript) has been used as a strong baseline in many works, including [1,2,3,4,5], to name a few. Notably, the benchmarking work of Duan et al. (COLM'24) identified that Ref performed the best on MIMIR. These facts support the validity of considering it as a competitive baseline, which is also why we particularly emphasized the comparison with Ref when discussing results on MIMIR. We believe being able to surpass or perform on par with the previously best method is a strong evidence that validates the effectiveness of our method.
>
>
> [1] An Empirical Analysis of Memorization in Fine-tuned Autoregressive Language Models
>
> [2] Did the Neurons Read your Book? Document-level Membership Inference for Large Language Models
>
> [3] Context-Aware Membership Inference Attacks against Pre-trained Large Language Models
>
> [4] Semantic Membership Inference Attack against Large Language Models
>
> [5] Scaling Up Membership Inference: When and How Attacks Succeed on Large Language Models
>
>
> **Q1: Nuances in training data detection**
>
>
> We appreciate the reviewer’s insightful comments regarding the importance of considering more fine-grained evaluation settings, such as distinguishing tokens and addressing nuances within pre-training corpora. We agree that such settings are crucial for advancing the field and represent an important direction for future research. However, creating these settings and conducting further investigation would require not only significant independent research efforts but also a collective push from the community. While our work focuses on proposing a new method that achieves improved performance within existing benchmarks and evaluation settings, we view this as a complementary step toward addressing the broader challenges highlighted by the reviewer.
>
> ------------
>
> [2/2] of authors' response

---

> ### Comment · Reviewer_5hJP · 2024-11-26
> **Response to Rebuttal**
>
> Hi Authors, Thank you for taking the time to write this rebuttal! Recently, it's been a busy time for all of us, so I am sorry for the late response. Below, I will respond to each point in detail.
>
> > $\mu_{x_{<t}}$ is the mean of token likelihoods over the vocabulary when predicting the current token..
>
> Thank you for clarifying this. This is the fundamental concept we need to begin with, so I am glad for the clarification. This looks more reasonable now. However, I still feel this notation is somewhat strange, as $x_{<t}$ represents the previous token sequence. I believe there should be a notation for the conditional here.
>
> > Recall our theoretical insight of "training data forms or locates near a local maximum (along each input dimension)", meaning that the training input tends to have higher probability than other neighbor input values. Under the discrete categorical distribution of LLMs, the insight can anologously translate to that the training token tends to have higher probability relative to many other candidate tokens in the vocabulary.
>
> Thank you for adding a paragraph in your revised version for this. However, I feel this connection is still somewhat weak. The input dimension from the ISM object corresponds to the hidden dimension, which has clear evidence from gradient descent. However, the conclusion you reached is "the training input tends to have higher probability than other neighbor input values." This conclusion is somewhat too shallow, and we don't need the ISM object section to derive something like "trained examples have high perplexity." For example, if you observe patterns from the final layer token hidden state (before the decoding head), this would be a natural transition.
>
> >  Yet, we believe that our method is more effective than existing methods in addressing this challenge, given the concrete example discussed with Figure 3. Specifically, when the training text is rare or difficult-to-learning, the predicted distribution across the vocabulary could be, say, near uniform, rendering the probability ...
>
> Thank you for adding this explanation. I feel this argument is reasonable and make sense to me.
>
> > Indeed, as the attention on this problem rapidly grows, we ourselves also see more works coming out recently. However, we clarify that at the time when we conducted our work, Min-K% was still one of the few works that targets LLM training data detection and was the state-of-the-art. In fact, our work is among the first that improve upon Min-K% and is prior to many recent works, including the mentioned arXiv preprint ReCall ..
>
> Lol this is something we may sadly have to accept in academia—I am not a harsh reviewer pushing authors to chase each other like this. However, I am raising this concern because the effectiveness of an MIA method is very important. When I notice that there is some work showing significant improvement compared to this work, it will influence my judgment for this round's submission.
>
> And I don't have further comments on W4 and Q1.
>
> Overall, I believe this is a great work with a good academic structure. I will slightly increase my score to reflect the rebuttal provided by the authors. Thank you for your time spent during this period!

---

> ### Author Response · Authors · 2024-11-26
> **Thank you**
>
> We sincerely thank the reviewer for sharing your detailed thoughts and giving a positive overall recommendation. We appreciate such in-depth discussion, which again helps us refine our thinking and make a better version of our work. Specifically regarding the notation, we have followed your suggestion to update the old notation of ($\mu_{x_{<t}}$, $\sigma_{x_{<t}}$) to ($\mu_{\cdot|x _ {<t}}$, $\sigma_{\cdot|x _ {<t}}$), which we think can make it clear that $x_{<t}$ is the condition, so that there is no ambiguity/confusion.

---

### Official Review · Reviewer_xYYm · 2024-11-03

**Soundness:** 4
**Presentation:** 3
**Contribution:** 3
**Rating:** 8
**Confidence:** 4

**Summary:**

This paper presents a method, Min-K%++, designed to detect pre-training data in LLMs by identifying local maxima in the model’s conditional distributions. The proposed approach theoretically grounds pre-training data detection in maximum likelihood training, showing that training instances often exhibit distinct distributional patterns. From the empirical experiments, it demonstrate that Min-K%++ achieving competitive performance on benchmarks like WikiMIA and MIMIR, with notable improvements over prior methods, particularly in challenging settings.

**Strengths:**

The originality of Min-K%++ lies in its focus on identifying local maxima to detect training data, moving beyond heuristic-based methods by leveraging theoretical insights from maximum likelihood training. The quality of the experiments is high, covering multiple benchmarks and various model sizes, showing consistent improvements over state-of-the-art baselines. The clarity is adequate in explaining the methodology and motivation, though some sections could benefit from clearer illustrations. The significance is evident, as pre-training data detection can enhance model transparency and accountability, which are critical in LLM applications.

**Weaknesses:**

The novelty of Min-K%++ is somewhat limited, as it builds incrementally on the existing Min-K% methodology rather than introducing a transformative approach. Additionally, the paper does not provide sufficient information about the computational resources required for implementing Min-K%++, which makes it difficult to assess the method’s feasibility on different hardware configurations or scales. While the benchmarks are well-chosen, a comparison with broader language tasks would better illustrate Min-K%++'s general applicability.

**Questions:**

- What is the computational overhead of Min-K%++ compared to baseline methods, and is it feasible for real-time applications?

- Would Min-K%++ perform well on diverse language corpora with complex semantics or high paraphrasing, or would adjustments be needed?

- Could you explain the theoretical choice of $\mu_{x<t}$ and $\sigma_{x<t}$ as calibration factors, especially in improving detection sensitivity in noisy settings?

---

> ### Author Response · Authors · 2024-11-20
>
> We very much thank the reviewer for giving us constructive feedbacks and acknowledging various aspects of our work, including originality, experiment quality, clarity, and significance. Below we address your comments/questions.
>
>
> **W1: Novelty**
>
> We totally understand that Min-K%++ might seem incremental w.r.t. Min-K% if only looking at their formulation. Yet, we'd also like to remark that the formulations of the two methods are derived from completely different paths: While Min-K% is designed solely based on heuristics, our method is developed with a solid theoretical interpretation/motivation, which is one of our novel and concrete contributions to the field.
>
>
> **W2: Lack of information on computational complexity**
>
> Thank you for pointing this out; indeed such information is important. Please see our detailed response to **Q1**. We have now included this information in Appendix A in the revised manuscript (marked in blue).
>
>
> **W3: Broader language tasks**
>
> We agree that evaluating with more benchmarks would better demonstrate the applicability of Min-K%++. Here we evaluate on a later released benchmark from Meeus et al. [1] which contains book texts. The benchmark is applicable to OpenLLaMA models [2], and below we show the detection AUROC on OpenLLaMA-7B and OpenLLaMA-13B. We see that Min-K%\++ still achieves superior results over competitive baselines.
>
> |           | OpenLLaMA-7B | OpenLLaMA-13B |
> |-----------|:------------:|:-------------:|
> | Loss      |     54.6     |      54.9     |
> | Ref       |     55.6     |      53.3     |
> | Lowercase |     54.7     |      53.3     |
> | Zlib      |     54.7     |      55.0     |
> | Neighbor  |     55.3     |      55.5     |
> | Min-K%    |     56.9     |      56.7     |
> | Min-K%++  |     **60.0**     |      **60.1**     |
>
>
> [1] Did the Neurons Read your Book? Document-level Membership Inference for Large Language Models
>
> [2] OpenLLaMA: An Open Reproduction of LLaMA
>
>
> **Q1: Computational complexity**
>
> Min-K%++ incurs no computational overhead and is as fast as the most lightweight baseline. We show the following python and pytorch-style code to illustrate this point.
>
> ```python
> # T: input token sequence length
> # V: vocabulary size of the model
> # input_ids [T,]: input token sequence
> # logits [T, V]: raw outputs of the model
> # probs [T, V]: softmax probability distribution over the vocabulary at each token position
> # log_probs [T, V]: log softmax probabilities over the vocabulary at each token position
> # token_log_probs [T,]: the log probability of each input token
>
> ########################################################
> # standard LLM inference; necessary for all methods
> logits = model(input_ids, labels=input_ids)
> probs = torch.nn.functional.softmax(logits[0, :-1], dim=-1)
> log_probs = torch.nn.functional.log_softmax(logits[0, :-1], dim=-1)
> token_log_probs = log_probs.gather(dim=-1, index=input_ids).squeeze(-1)
>
> ########################################################
> # operations specific to Min-K%++
> # basic algebraic operations with negligible computational cost
> mu = (probs * log_probs).sum(-1)
> sigma_square = (probs * torch.square(log_probs)).sum(-1) - torch.square(mu)
> # Equation (3)
> mink_pp = (token_log_probs - mu) / sigma_square.sqrt()
> # Equation (4)
> mink_pp_aggregated = numpy.mean(np.sort(mink_pp)[:int(len(mink_pp * k))])
> ```
>
> Concretely, the computational cost almost solely comes from the one-time LLM inference itself, which is shared by all methods. After obtaining the output statistics, the detection score of Min-K%++ are computed with basic algebraic operations, whose computational overhead is negligible especially compared with LLM inference.
>
>
> In terms of comparison with baseline methods, Min-K%++ is as fast as the fastest ones (including Loss, Zlib, and Min-K%), as their execution all consists of one forward pass of the LLM and then some basic algebraic operations. Ref, Lowercase, and Neighbor all requires multiple forward passes of the LLM, thus being significantly slower than the other methods.
>
> -----------------
> [1/2] of authors' response

---

> ### Author Response · Authors · 2024-11-20
>
> **Q2: Complex semantics or high paraphrasing**
>
> Thank you for this question, which we believe is a crucial problem to think as the field continues to progress. For texts with complex semantics, we don't think Min-K%++ would need adjustments because its underlying mechanism makes no assumption on the actual semantics. Meanwhile, notice that the used MIMIR benchmark already consists of diverse, real-world texts from various domains including ArXiv, PubMed, and DM Mathematics etc, on which our Min-K%++ achieves superior or competitive performance. In terms of paraphrasing, given the improved results yielded by our method on paraphrased Wikipedia, we have reason to anticipate that Min-K%++ can remain more effective than others in high paraphrasing settings with no or minimum changes.
>
>
> **Q3: Calibration factors**
>
>
> Thank you for raising this question, which greatly helps us refine our thinking and clarify these ideas. Please see our answer below; the discussion is also updated in Section 4.2 for clarity.
>
>
> For $\mu_{x_{<t}}$, recall our theoretical insight of "training data forms or locates near a local maximum" (i.e., training input tends to have higher probability than other neighbor input values). Under the discrete categorical distribution of LLMs, the insight can anologously translate to that training token tends to have higher probability relative to other candidate tokens within the vocabulary. Our formulation naturally reflects this idea by comparing $\log p(x_t|x_{<t})$ (log probability of the input token) with $\mu_{x<t}$ (representative for the probability of other candidate tokens).
>
>
> The reason that having $\mu_{x_{<t}}$ could be effective especially in noisy settings is best illustrated with the example in Figure 3. When for certain reason the predicted distribution across the vocabulary is near uniform (thus the probability of the training token $\log p(x_t|x_{<t})$ is low), existing methods like Min-K% may as a result mark it as non-training since they simply examine the absolute probability. In contrast, our method can still identify the relative largeness of $\log p(x_t|x_{<t})$ by measuring $\log p(x_t|x_{<t})-\mu_{x_{<t}}$, which in turn helps detect training data in such challenging and noisy scenarios.
>
>
> The use of $\sigma_{x_{<t}}$ is inspired by temperature scaling, a well-established method for calibrating the prediction probability of neural networks. Such technique is conceptually well-suited for our task since we are closely looking at the predicted probabilities of LLMs. In the meantime, while temperature scaling uses a constant scaling factor, we posit that for training data detection the suitable temperature varies across the input and the model. Therefore, we intend to have a dynamic scaling factor that is adaptive to different cases, which would make the calibrated score more robust. Specifically, by using the standard deviation $\sigma_{x_{<t}}$ as the calibration factor, we are essentially doing z-score normalization over $\log p(x_t|x_{<t})$ and making the value of $\log p(x_t|x_{<t})$ more comparable across different cases. Thus $\sigma_{x_{<t}}$ is expected to help achieve accurate and robust detection.
>
> --------
> [2/2] of authors' response

---

> > ### Comment · Reviewer_xYYm · 2024-11-24
> >
> > Thanks for the response and clarification, I have improved my rating.

---

> > > ### Author Response · Authors · 2024-11-26
> > > **Thank you**
> > >
> > > We sincerely thank the reviewer again for reviewing our work and giving us positive feedback. We greatly appreciate your insights and suggestions, which have helped improve our work in various aspects.

---

### Meta-Review · Area_Chair_mKLE · 2024-12-16

**Metareview:**

This paper presents a method Min-K%++ that builds on the previous Min-K% to detect pre-training data in LLMs by identifying local maxima in the model’s conditional distributions. In experiments, Min-K/%++ achieved competitive performance on benchmarks like WikiMIA and MIMIR. The paper is easy to follow. The method is well motivited and technically sound. The reviewers raised concerns about the novelty of the method, the evaluation and the presentation. After the author rebuttal, all the reviewers were positive about this paper. I'd like to point out a major issue with the datasets used in the evaluation: datasets based on cut-off dates such as WikiMIA can be heavily biased and thus cannot well validate the performance of the proposed method, as shown in recent studies (Duan et al., 2024/arXiv:2402.07841 and Maini et al., 2024/arXiv:2406.06443). Experimental results on addtional datasets need to be added in the next version.

**Additional Comments On Reviewer Discussion:**

The reviewers raised concerns about the novelty of the method, the evaluation and the presentation. After the author rebuttal,  three reviewers increased their scores and all of them were positive about this paper.

---

### Decision · Program_Chairs · 2025-01-22

Accept (Spotlight)